# Matrix stiffness induces midnolin-dependent lamin B1 degradation to control myoblast differentiation

Liping Guo[1,2,3,10], Yanjing Zhao[1,4,10], Zhe Zhang [ID][1], Chang Sun[1,5], Yafan Xie[6], Qin Dai[1,7,8], Yan Yan [ID][4], Yaoqi Zhou [ID][1], Yang Zhang [ID][9], Quhuan Li [ID][2✉], Juhui Qiu [ID][6✉] & Qin Peng [ID][1✉]

## Abstract

**Cells decode mechanical cues to direct fate decisions through nuclear remodeling, yet nuclear adaptors to mechanical signals remain elusive. Here, we show that soft matrix suppresses myoblast differentiation and induces nuclear abnormality within 30 min, accompanied by a greater than 60% reduction in lamin B1 proteins levels. Mechanistically, midnolin interacts with lamin B1 and mediates ubiquitination-independent degradation of lamin B1 on soft matrix, through the Catch domain of midnolin engaging a β-strand within lamin B1's Ig-like domain. Functionally, moderate lamin B1 expression is essential for myoblast differentiation initiation, as its depletion either by siRNA or CRISPR knockout abolishes myogenic capacity. Our findings reveal that the midnolin-proteasome axis directly converts mechanical inputs into lineage commitment by triggering lamin B1 degradation, defining a novel nuclear mechano-adaptation pathway.**

**Keywords** Matrix Stiffness; Lamin B1 Degradation; Midnolin-proteasome Pathway; Myoblast Differentiation
**Subject Categories** Cell Adhesion, Polarity & Cytoskeleton; Development; Post-translational Modifications & Proteolysis

## Introduction

Mechanical stimuli in the cellular microenvironment are powerful regulators of cell function and behavior, such as extracellular matrix (ECM) elasticity and mechanical strain can significantly influence the fate decision of stem and progenitor cells (Baghdadi et al, 2024; Shiraishi et al, 2023). The basic mechanism is that stem cells sense mechanical cues and then active downstream signaling events and ultimately results in cell fate transitions and even participates in diseases development (Fiore et al, 2025). For example, studies in satellite cells or the differentiation of muscle progenitor cells (Kjaer, 2004; Thomas et al, 2015; Zhang et al, 2021) revealed that the structural remodeling and stiffness of the ECM are involved in Duchenne muscular dystrophy (Long et al, 2024), both reduced mechanical loading and low ECM stiffness suppress cell differentiation and muscle regeneration, causing muscle disuse atrophy (Gibbons et al, 2018; Kjaer, 2004; Wall et al, 2013). Hence, the cellular and molecular mechanistic understanding of how cells sense mechanical cues are still the potential mechanical determinators of cell fate.

The nucleus, as the stiffest cellular organelle, perceives mechanical cues through cytoskeletal connections and the linker of nucleoskeleton and cytoskeleton (LINC) complex-mediated mechanotransduction (De Belly et al, 2022; Dupont and Wickstrom, 2022; Kalukula et al, 2022; Maurer and Lammerding, 2019; Nava et al, 2020). This mechanosensing capability enables dynamic modulation of nuclear morphology and transcriptional programs, ultimately influencing cell fate decisions. On stiff matrices, nuclei typically adopt a flattened and elongated morphology, which is associated with differentiated cell states. In contrast, on soft matrices, nuclei remain more rounded, exhibiting nuclear envelope (NE) wrinkling and reduced volume-a configuration that helps maintain stem cell pluripotency (Cosgrove et al, 2021; Lovett et al, 2013; Nguyen et al, 2024; Price et al, 2017; Virdi and Pethe, 2022). The proper adaptation from the nucleus to the extracellular mechanics is typically called nuclear mechano-adaptation, which requires the nuclear skeleton to rapidly adapt the changes to maintain nuclear integrity (Echarri et al, 2019). However, there are lacking a precise explanation of the temporal and molecular basis underpinning nuclear adaptation to the mechano-microenvironment (Beedle and Roca-Cusachs, 2023).

NE proteins maintain the structural integrity and stability of the nuclei. Loss of nuclear skeleton lamina proteins can cause nuclear envelope wrinkling and alterations in nuclear volume (Vahabikashi et al, 2022). Previous studies showed that loss of lamin A tends to increase nuclear volume while loss of lamin B1 behaves in the opposite way (Swift et al, 2013; Vahabikashi et al, 2022). Lamin B1 expression is responsible for nuclear elasticity to stabilize

[1]Institute of Systems and Physical Biology, Shenzhen Bay Laboratory, Shenzhen, China. [2]School of Biology and Biological Engineering, South China University of Technology, Guangzhou, China. [3]School of Life Science and Technology, Harbin Institute of Technology, Harbin, China. [4]Division of Life Science, Hong Kong University of Science and Technology, Clear Water Bay, Hong Kong, China. [5]Department of Pathology and Experimental Therapeutics, Faculty of Medicine and Health Sciences, Barcelona University, Barcelona, Spain. [6]Key Laboratory for Biorheological Science and Technology of Ministry of Education, State and Local Joint Engineering Laboratory for Vascular Implants, College of Bioengineering, Chongqing University, Chongqing, China. [7]Shenzhen Medical Academy of Research and Translation (SMART), Shenzhen, China. [8]Westlake University, Hangzhou, China. [9]Institute of Molecular Physiology, Shenzhen Bay Laboratory, Shenzhen, China. [10]These authors contributed equally: Liping Guo, Yanjing Zhao.
✉E-mail: liqh@scut.edu.cn; jhqiu@cqu.edu.cn; pengqin@szbl.ac.cn

chromatin condensation (Wintner et al, 2020). However, the timescales and mechanism on how lamin B1 loss adapts to the mechano-microenvironment is not well understood. Currently, there are three pathways related to protein loss or degradation, which are ubiquitination-dependent proteasome pathway (Pohl and Dikic, 2019), midnolin-proteasome pathway for ubiquitination-independent degradation (Gu et al, 2023), as well as autophagy-lysosomal pathway (Pohl and Dikic, 2019). It is reported that lamin B1 undergoes degradation through ubiquitination-regulated pathways by E3 ligases (Khanna et al, 2018; Krishnamoorthy et al, 2018), or autophagic degradation after oncogenic damage (Dou et al, 2015). The mechanism underlying ECM stiffness-dependent regulation of lamin B1 proteostasis remains unclear.

In this study, we observed the significant reduction of myoblast differentiation on soft matrices, which was attributed to time-dependent nuclear abnormalities. We analyzed that the major change of nuclear membrane proteins was lamin B1 reduction on soft matrix. To elucidate the underlying mechanism, we investigated different pathways that regulate proteostasis and identified that midnolin mediated lamin B1 degradation via a β-strand capture mechanism in the ubiquitination-independent proteasome pathway on soft matrix. Furthermore, we explored the consequences of lamin B1 reduction in myoblast differentiation. We found that lamin B1 upregulation is required in the early stages of differentiation and influences key myoblast differentiation regulators: *Myod1* and *Wnt4*. Our findings suggest that lamin B1 serves as a critical regulator of nuclear mechanoadaptation and is indispensable for proper myoblast differentiation.

## Results

### Soft Matrix attenuates myoblast differentiation related to nuclear abnormalities

To explore the effect of mechanical cues on muscle cell differentiation, mouse myoblasts were cultured on polyacrylamide (PAA) hydrogels with stiffnesses of 0.2 kPa (soft), mimicking atrophic muscle conditions, and 10 kPa (stiff), resembling healthy muscle stiffness (Figs. 1A and EV1A). RNA sequencing (RNA-seq) categorized the biological process of differentially expressed genes (DEGs) by Gene Ontology (GO) analysis, revealing significant alternations in cell fate and differentiation catalogers under soft matrix condition (Fig. 1B).

We further induced primary myoblast differentiation with 2% horse serum on different stiffness matrix. Our results demonstrated that myoblast fusion, as indicated by myosin heavy chain 4 (MYH4) expression, was markedly suppressed on soft matrix relative to stiff matrix following 7 days of differentiation induction (Fig. 1C–E). Similar results were from myoblast cell line C2C12 after 7 days induction (Fig. 1F–H), and this inhibition was evident as early as day 3 (Fig. EV1B,C).

RNA-seq also indicated the enrichment of the pathway related to DNA repair (Fig. 1B), suggesting that the impaired myoblast differentiation might be associated with the accumulation of DNA damage. To test this hypothesis, we checked DNA damage dynamics in cells cultured on matrix of varying stiffness during the initial 72-h differentiation period. More phosphorylated H2AX at Ser139 (γH2AX)-positive nuclei were detected on soft matrix (Fig. 1I–K), demonstrating that soft matrix caused more DNA damage and may subsequently affected myoblast differentiation. Furthermore, nuclear abnormalities formed prior to a bulk accumulation of γH2AX. Immunostaining revealed a significant alteration of nuclear morphology in 1 Day (Fig. 1M). In particular, 16.7% C2C12 cells cultured on soft matrix displayed a marked increase in nuclear blebbing and micronuclei (MN) formation (Fig. 1L).

In addition, immunostaining results showed that soft matrix hindered F-actin assembly in the early stage of cell adhesion (0.5 h) followed by gradual spreading (3 h) in C2C12 (Fig. 1N,O). Nucleus, as a mechanosensor, responds to external forces transmitted from the cytoskeleton and deforms accordingly (Kalukula et al, 2022). Our results showed a significant reduction in nuclear volume on soft matrix, compared to stiff matrix (Fig. 1P).

Taken together, myoblast differentiation is inhibited when myoblasts cultured on soft matrix. Our findings indicate that this inhibition involves a time-dependent cellular adaptation process. Initially, within several minutes, cells on soft matrix exhibited reduced spreading capacity and nuclear volume. Subsequently, within several hours, nuclear morphological abnormalities emerged, subsequently triggering γH2AX foci formation as a marker of DNA damage response. These sequential events collectively contribute to the suppression of myoblast differentiation over extended culture periods (Fig. 1Q).

### Lamin B1 decreases with reduced nuclear volume on soft matrix

Nucleus abnormalities are influenced by NE proteins such as lamin A/C and lamin B1, which serve as nuclear skeleton with mechanical properties (Vahabikashi et al, 2022). Therefore, we were curious about how nuclear envelop proteins are responsible for maintaining nucleus morphology and volume on matrix. Here we screened the level changes of the majority of NE proteins that have been reported to be associated with mechano-signaling pathways (Donnaloja et al, 2019; Kalukula et al, 2022) by western blot, including Nucleoporin 153 (NUP153), SUN1/2, lamin B receptor (LBR), emerin, lamina associated protein 2β (LAP2β), lamin A/C, lamin B1 (Fig. 2A). Our western blot results showed that the protein levels of lamin A/C was lower on soft matrix as previously reported (Swift et al, 2013), meanwhile both lamin B1 and LAP2β decreased significantly on soft matrix (Figs. 2B and EV2A,B). Based on the quantitative analysis, reduction of lamin B1 protein level on soft matrix was the most among all the NE proteins at 0.5 h post cell seeding (Fig. 2B,C). Additionally, lamin B1, which is critical for myoblast differentiation, represents a novel mechanism in nuclear mechano-adaptation. Therefore, we selected lamin B1 as a biomarker to investigate nuclear mechanosensing dysregulation in muscle atrophy. Immunostaining further confirmed that lamin B1 protein level reduced on soft matrix (Fig. 2D,E). This reduction was similarly observed in primary myoblasts (Fig. 2F,G), confirming the stiffness-dependent regulation of lamin B1. Next, we detected lamin B1 protein levels over a day and found that the low lamin B1 level on soft matrix was retained over the time from 0.5 h to 24 h (Fig. 2H,I). Consistent reduction of lamin B1 protein levels on soft matrix was observed in both whole-cell lysates and nuclear fractions (Fig. EV2C–E). Altogether, these results implicate that

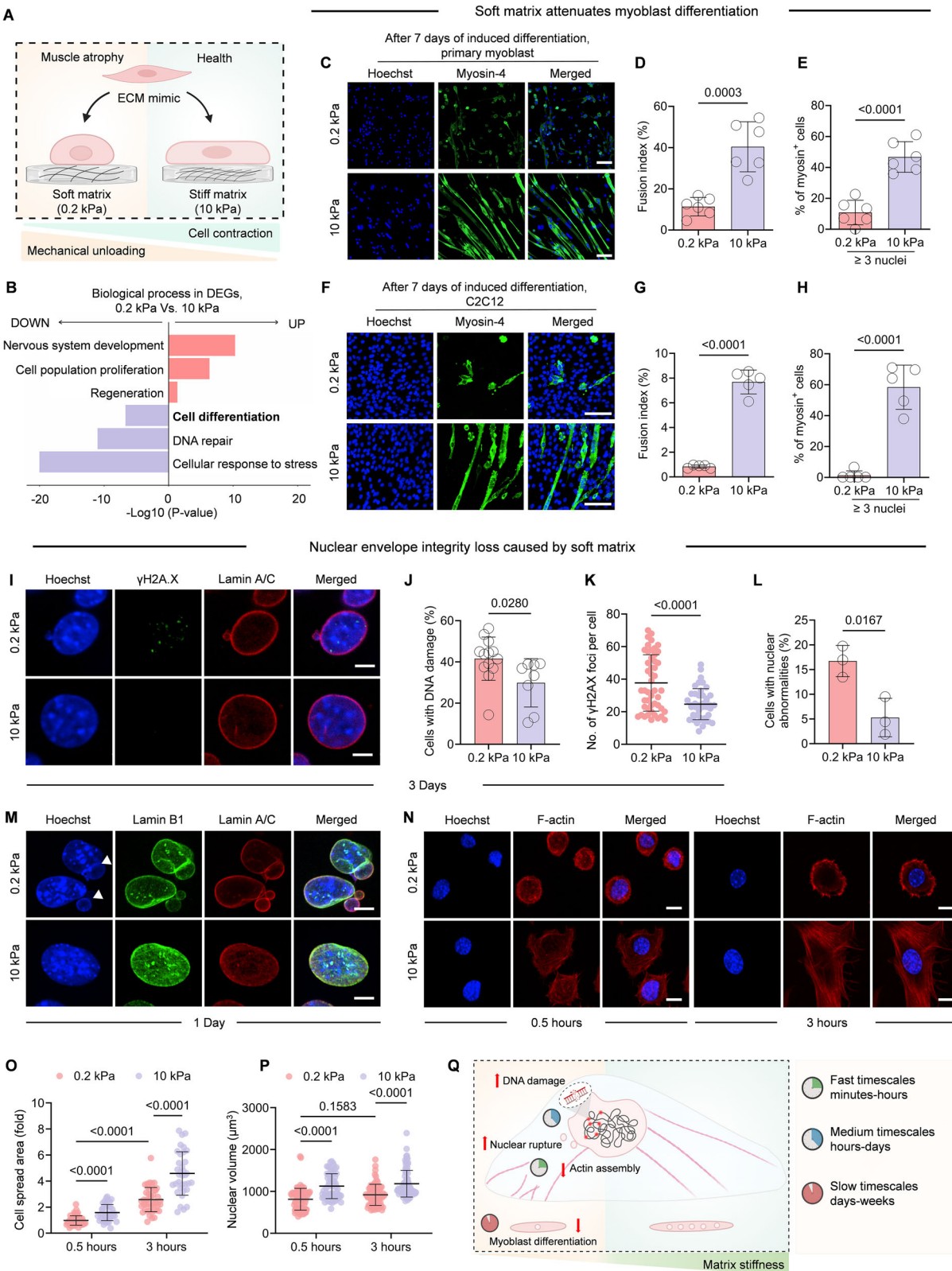

**A** Muscle atrophy — Health
ECM mimic
Soft matrix (0.2 kPa) — Stiff matrix (10 kPa)
Mechanical unloading — Cell contraction

**B** Biological process in DEGs, 0.2 kPa Vs. 10 kPa
DOWN ← → UP
Nervous system development
Cell population proliferation
Regeneration
**Cell differentiation**
DNA repair
Cellular response to stress
-Log10 (P-value)

Soft matrix attenuates myoblast differentiation

**C** After 7 days of induced differentiation, primary myoblast
Hoechst | Myosin-4 | Merged
0.2 kPa
10 kPa

**D** Fusion index (%) 0.0003 — 0.2 kPa, 10 kPa

**E** % of myosin+ cells <0.0001 — 0.2 kPa, 10 kPa ≥ 3 nuclei

**F** After 7 days of induced differentiation, C2C12
Hoechst | Myosin-4 | Merged
0.2 kPa
10 kPa

**G** Fusion index (%) <0.0001 — 0.2 kPa, 10 kPa

**H** % of myosin+ cells <0.0001 — 0.2 kPa, 10 kPa ≥ 3 nuclei

Nuclear envelope integrity loss caused by soft matrix

**I** Hoechst | γH2A.X | Lamin A/C | Merged
0.2 kPa
10 kPa

**J** Cells with DNA damage (%) 0.0280 — 0.2 kPa, 10 kPa

**K** No. of γH2AX foci per cell <0.0001 — 0.2 kPa, 10 kPa

**L** Cells with nuclear abnormalities (%) 0.0167 — 0.2 kPa, 10 kPa

3 Days

**M** Hoechst | Lamin B1 | Lamin A/C | Merged
0.2 kPa
10 kPa
1 Day

**N** Hoechst | F-actin | Merged | Hoechst | F-actin | Merged
0.2 kPa
10 kPa
0.5 hours — 3 hours

**O** Cell spread area (fold) 0.2 kPa, 10 kPa
<0.0001, <0.0001, <0.0001
0.5 hours, 3 hours

**P** Nuclear volume (µm³) 0.2 kPa, 10 kPa
<0.0001, 0.1583, <0.0001
0.5 hours, 3 hours

**Q** DNA damage
Nuclear rupture
Actin assembly
Myoblast differentiation
Fast timescales minutes-hours
Medium timescales hours-days
Slow timescales days-weeks
Matrix stiffness

**Figure 1. Soft matrix impairs myoblast differentiation and increases nuclear abnormalities and DNA damage.**

(A) Schematic of hydro-matrices with different stiffness to mimic mechanical loading on cells in healthy and muscular dystrophy conditions (figure created with BioRender.com). (B) GO terms analysis of biological processes for upregulated and downregulated genes (0.2 kPa vs. 10 kPa) after 12 h of culture on matrices. Differential expression was analyzed using the Wald test in DESeq2; |log₂FC| > 1, adjusted $P < 0.05$. (C–H) Representative images of differentiated mouse primary myoblast cells (C) and C2C12 cells (F) on fibronectin (FN)-coated PAA matrices after 7 days of 2% horse serum induction. scale bar, 100 μm. (D, E, G, H) Quantification data of the fusion index and the percentage of myosin⁺ cells that ≥3 nuclei in (C, F), respectively. $n = 7$ (D, E), 5 (G, H) biological replicates. Statistical significance: (E) $P = 4.10 \times 10^{-5}$, (G) $P = 2.74 \times 10^{-7}$, (H) $P = 2.29 \times 10^{-5}$. (I–K) Representative images from C2C12 cells seeded on FN-coated PAA matrices after 3 days of horse serum induction. Hoechst for nuclear staining in blue, γH2AX staining in green and lamin A/C staining in red. Scale bars, 5 μm. (J, K) The Quantification of cell numbers with DNA damage (J) and γH2AX foci (K) in (I). $n > 40$ cells for each condition. $P = 3.92 \times 10^{-5}$. (L) Quantification of cells with nuclear abnormalities in (M). (M) Representative images from C2C12 cells seeded on FN-coated PAA matrices after 1 days of horse serum induction. Hoechst for nuclear staining in blue, lamin B1 staining in green and lamin A/C staining in red. White arrows indicate nuclear blebbing and micronuclei formation. Scale bars, 5 μm. $n = 3$ biological replicates. (N) Immunofluorescence of C2C12 cells seeded onto FN and gelatin-coated PAA matrices at 0.5 h and 3 h, respectively. Hoechst for nuclear staining in blue and F-actin staining in red. Scale bars, 10 μm. (O, P) Fold changes of cell spread area (O) and nuclear volume (P) were quantified from (N). $n > 50$ cells for each group. Data are presented as the mean ± SD. Tukey's multiple comparisons test. In (O), $P$ values are $2.32 \times 10^{-5}$, $1 \times 10^{-10}$, and $8.08 \times 10^{-7}$, respectively. In (P), $P$ values are $3.88 \times 10^{-8}$, 0.1583, and $6.36 \times 10^{-8}$, respectively. (Q) Summary of hypothesis for soft matrix - mediated inhibition of myoblast differentiation (figure created with BioRender.com). Data information: data in (D, E, G, H, J, K, L) were presented as the mean ± SD. Two-tailed Student's $t$ test was used for statistical analysis. Source data are available online for this figure.

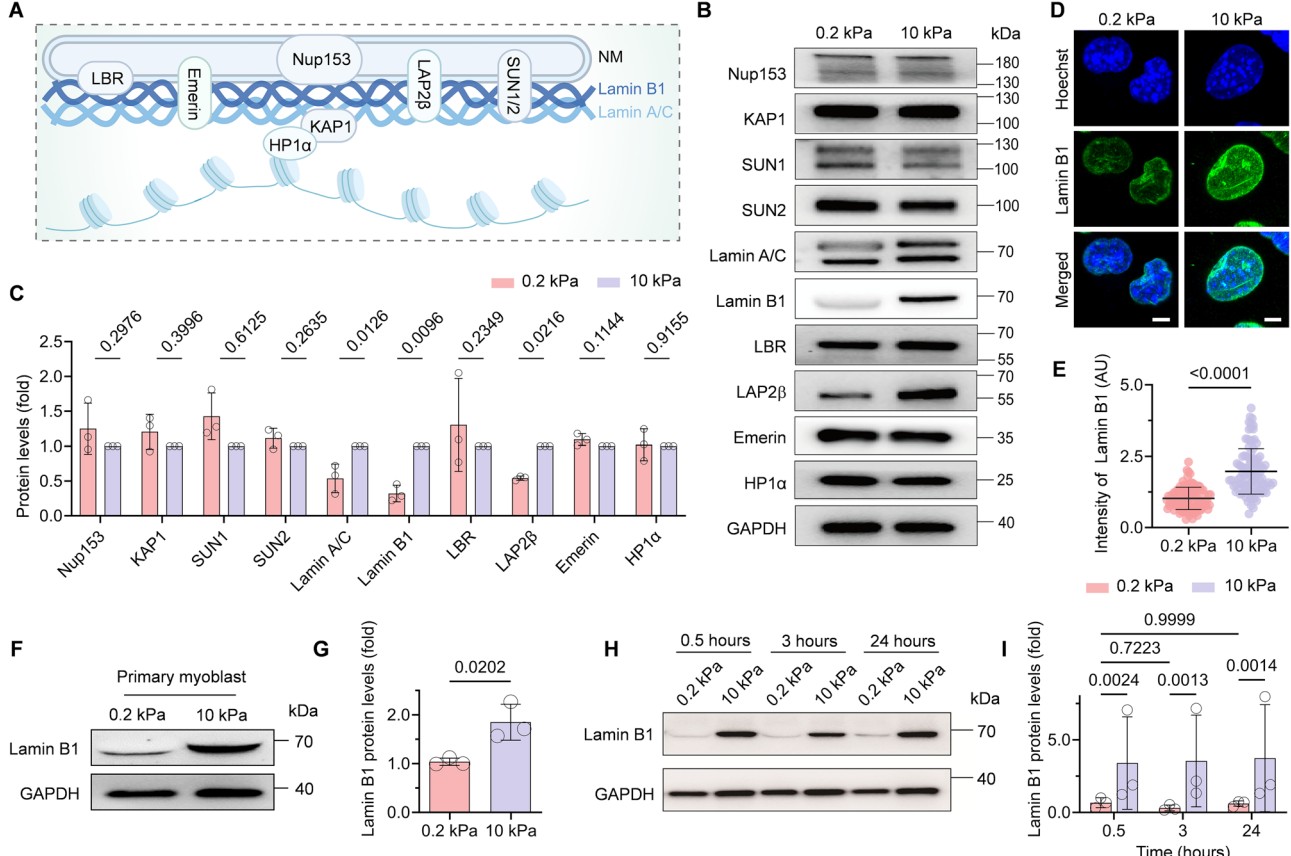

**Figure 2. Soft matrix induces a significant reduction in lamin B1 and maintains this difference over the time consistently.**

(A) Schematic diagram of some NE proteins (figure created with BioRender.com). (B, C) Western blot analysis of NE proteins displayed in (A) from C2C12 seeding onto FN and gelatin-coated PAA matrices after 0.5 h. $n = 3$ biological replicates, Data are presented as the mean ± SD. Two-tailed Student's $t$ test. (D) Immunofluorescence of C2C12 cells on FN and gelatin-coated PAA matrices. Hoechst for nuclear staining in blue and lamin B1 staining in green. Scale bars, 4 mm. (E) Quantitative analysis of fluorescence intensity of lamin B1 on 0.2 kPa and 10 kPa substrates from (D) (>300 cells for each group, Two-tailed Student's $t$ test, Data are presented as the mean ± SD). $P = 1 \times 10^{-10}$. The fluorescence intensity unit is arbitrary unit (AU) and is defined from the detected total fluorescence intensity normalized. (F, G) Western blot analysis of lamin B1 proteins in mouse primary myoblast cells seeded onto FN-coated PAA matrices after 0.5 h. $n = 3$ biological replicates. Two-tailed Student's $t$ test, Data are presented as the mean ± SD. (H, I) Western blot analysis of lamin B1 proteins in C2C12 cells seeded onto FN and gelatin-coated PAA matrices at different times. $n = 3$ biological replicates. Two-way ANOVA/Tukey's multiple comparisons test. Data are presented as the mean ± SD. Source data are available online for this figure.

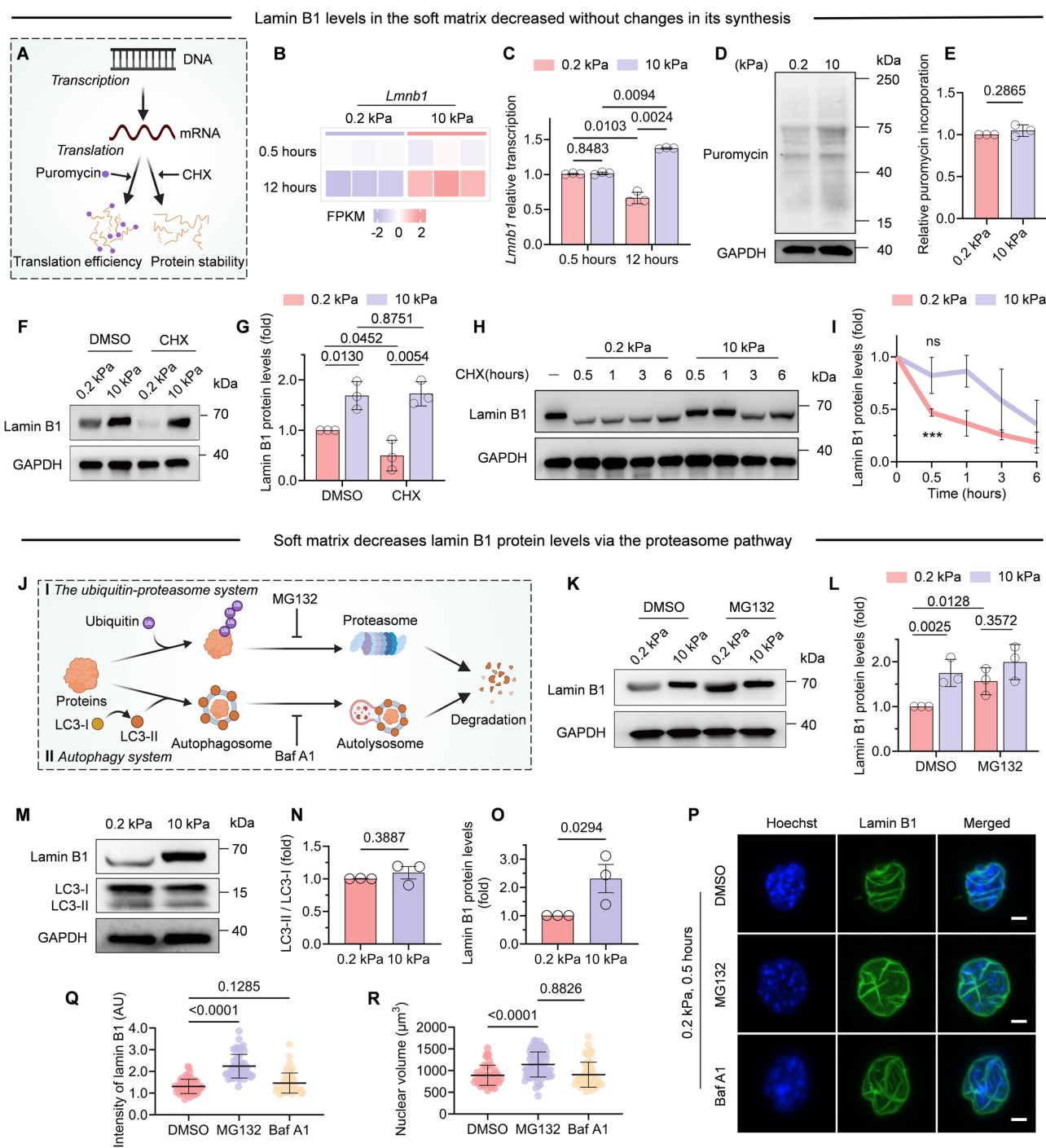

matrix stiffness modulates the physical deformation of nucleus via modulation of lamin B1 protein level.

## Lamin B1 is degraded by proteosome pathway on soft matrix

Cells maintain proteostasis by dynamically balancing protein synthesis and degradation in response to external stimuli. To elucidate the mechanism underlying the differential lamin B1 expression observed between 0.2 kPa and 10 kPa matrix, we

investigated whether this regulation was mediated through changes in protein synthesis or altered degradation kinetics (Fig. 3A).

To investigate the influence of matrix stiffness on lamin B1 protein synthesis, firstly we performed RNA sequencing analysis of C2C12 myoblasts cultured on matrix of varying stiffness at two critical timepoints (0.5 h and 12 h) post-seeding. RNA-seq analysis demonstrated comparable mRNA expression levels of *Lmnb1* between soft and stiff matrix at 0.5 h. However, by 12 h, stiff matrix exhibited significantly elevated *Lmnb1* transcript levels. These findings indicate that the rapid decline in lamin B1 protein

◀ **Figure 3. Lamin B1 degradation on soft matrix through proteasome pathway.**

(A) Schematic diagram of classical protein synthesis pathway. Puromycin is incorporated into nascent polypeptides to detect protein synthesis and assess translation efficiency. Cycloheximide (CHX) inhibits ribosomal activity to block new protein synthesis, enabling tracking of pre-existing protein decay for stability analysis. (Created with BioRender.com). (B) Representative genes from RNA-seq were upregulated in pink and downregulated in violet after 0.5 h and 12 h culturing on FN and gelatin-coated PAA matrices (0.2 kPa Vs. 10 kPa). (C) The expression of *Lmnb1* from RNA-seq after 0.5 h and 12 h culturing on FN and gelatin-coated PAA matrices (0.2 kPa Vs. 10 kPa). $n = 3$ biological replicates. (D, E) C2C12 were treated with 91 μM puromycin for 5 min after seeding onto FN-coated PAA matrices for 25 min and immunoblotted for puromycin. $n = 3$ biological replicates. (F, G) Western blot analysis of lamin B1 protein in C2C12 seeded onto FN and gelatin-coated PAA matrices for 0.5 h in the presence of protein synthesis inhibitor (CHX, 50 mg/mL). $n = 3$ biological replicates. (H, I) Western blot analysis of lamin B1 proteins stability in C2C12 before seeding (−) and post-seeding onto FN-coated PAA matrices under cycloheximide (CHX) treatment to inhibit protein translation. $n = 3$ biological replicates. At 0.5 h, compared to the condition before seeding ($t = 0$), lamin B1 protein level showed a significant decrease on 0.2 kPa gel (\*\*\*$P = 0.0001$), while no significant change was observed on 10 kPa gel (ns, no significance, $P = 0.3616$). Data are presented as the mean ± SEM, with two-tailed Student's $t$ test. (J) Schematic diagram of two classical protein degradation pathways. Treatment with MG132 inhibits proteasome function by blocking its proteolytic activity, and Baf A1 (bafilomycin A1) inhibits autophagy by inhibiting the fusion between autophagosomes and lysosomes (created with BioRender.com). (K, L) Western blot analysis of lamin B1 protein in C2C12 seeded onto FN and gelatin-coated PAA matrices for 0.5 h in the presence of MG132 (10 μM). $n = 3$ biological replicates. (M-O) Western blot analysis of lamin B1 protein and LC3-II/LC3-I protein in C2C12 seeded onto FN-coated PAA matrices for 0.5 h in the presence of Baf A1 (100 nM). $n = 3$ biological replicates. (P) Immunofluorescence of C2C12 cells on FN and gelatin-coated 0.2 kPa substrates for 0.5 h with DMSO or MG132 or Baf A1 treatment. Hoechst for nuclear staining in blue and lamin B1 staining in green. Scale bars, 4 μm. (Q, R) Quantitative analysis of the fluorescence intensity of lamin B1 (Q) and nuclear volume (R) for the images from (P) (>50 cells for each condition, one-way ANOVA/Dunnett's multiple comparisons test. Data are presented as the mean ± SD. In (Q), $P$ values are $1 \times 10^{-10}$, and 0.1285, respectively. In (R), $P$ values are $1.11 \times 10^{-6}$ and 0.8826, respectively. Data information: in (C, E, G, L, N, O) as the mean ± SD, with two-tailed Student's $t$ test. Source data are available online for this figure.

observed on soft matrix (Fig. 2B) is mediated through post-transcriptional regulation rather than reduced gene expression (Fig. 3B,C). Secondly, to determine whether translation was mediated by matrix stiffness, we confirmed by puromycin incorporation that global protein synthesis is unaffected by matrix stiffness (Fig. 3D,E), suggesting that matrix stiffness does not affect protein translation in adhesion process.

C2C12 myoblasts were then plated onto soft or stiff matrices in the presence or absence of cycloheximide (CHX), a selective inhibitor of translational elongation (Fig. 3A). CHX was added to the cell suspension before seeding. Our results showed that, lamin B1 protein level decreased significantly on soft matrix after 0.5 h of CHX treatment compared to DMSO group (Fig. 3F,G), whereas no change was observed on stiff matrix. To measure the kinetics of lamin B1 protein degradation when cells were exposed to soft and stiff matrix, we treated C2C12 cells with CHX at different time points to assess the stability of lamin B1. Our results showed that B1 exhibited degradation kinetics on 0.2 kPa matrix, with significant protein loss detectable within 30 min of treatment. In contrast, lamin B1 remained stable for at least 60 min on 10 kPa matrix (Fig. 3H,I), demonstrating stiffness-dependent regulation of lamin B1 proteostasis.

Collectively, our findings demonstrate that accelerated protein degradation serves as the primary mechanism underlying the depletion of lamin B1 on soft matrix. Protein degradation occurs primarily through two evolutionarily conserved pathways: the ubiquitin-proteasome system and autophagy-lysosome pathway, which collectively mediate the controlled breakdown of proteins into reusable amino acids and short peptides (Fig. 3J) (Balchin et al, 2016; Pohl and Dikic, 2019). Inhibition of proteasomal degradation using MG132 attenuated the decrease in lamin B1 protein level on soft matrix, while there was no effect on the cells on stiff matrix (Fig. 3K,L). A similar trend was observed upon treatment with bafilomycin A1 (Baf A1), which blocks autophagosome–lysosome fusion (Fig EV3A,B), suggesting that multiple degradation pathways may be involved under soft matrix. To examine potential autophagic degradation, we monitored LC3 lipidation, a marker of autophagy pathway activation (Fig. 3J). Quantitative analysis revealed no significant difference in the LC3-II/LC3-I ratio between

soft and stiff matrix (Fig. 3M–O). Immunofluorescence results further confirmed lamin B1 protein level and nuclear volume was significantly recovered with MG132 treatment but not Baf A1 (Fig. 3P–R). These results demonstrate the proteasome-dominant degradation of lamin B1 protein is regulated by matrix stiffness.

## Midnolin-proteasome pathway mediates lamin B1 degradation on soft matrix

Given that conventional protein degradation pathways rely on ubiquitination (Pohl and Dikic, 2019), we sought to investigate the mechanoadaptation mechanism underlying lamin B1 degradation on soft matrix. First, we treated C2C12 cells with MG132 and conducted co-immunoprecipitation (Co-IP) analysis. No substantial ubiquitination on lamin B1 protein was detected on soft matrix (Fig. EV4A), implying lamin B1 degradation occurs through ubiquitination-independent proteasomal pathway.

Latest study identified a novel ubiquitin-independent proteasomal degradation pathway mediated by the nuclear protein midnolin (Gu et al, 2023). Midnolin captures proteins through its Catch domain and guides them to proteasomal degradation (Fig. 4A). To determine whether midnolin contributes to lamin B1 degradation, we firstly checked midnolin levels on different stiffness matrix. Our results revealed a significant decrease in midnolin protein levels in C2C12 cells cultured on 0.2 kPa matrix compare to 10 kPa matrix within 30 min of seeding. This autoregulatory reduction in midnolin abundance on soft matrix mirrors the mechanosensitive degradation kinetics observed for lamin B1 (Fig. 4B,C). Midnolin knockdown by siRNA significantly elevated lamin B1 levels on soft matrix but no significant effect on stiff matrix, demonstrating that midnolin-mediated lamin B1 degradation is stiffness-dependent (Figs. 4D,E and EV4B). Immunofluorescence results further confirmed a twofold increase in lamin B1 levels on soft matrix upon midnolin knockdown compared to the control group (Fig. 4F,G). However, midnolin knockdown did not prevent nuclear volume reduction (Fig. 4F,H), implying other mechano-sensitive nuclear morphology involves midnolin-independent pathways, potentially mediated by other NE components.

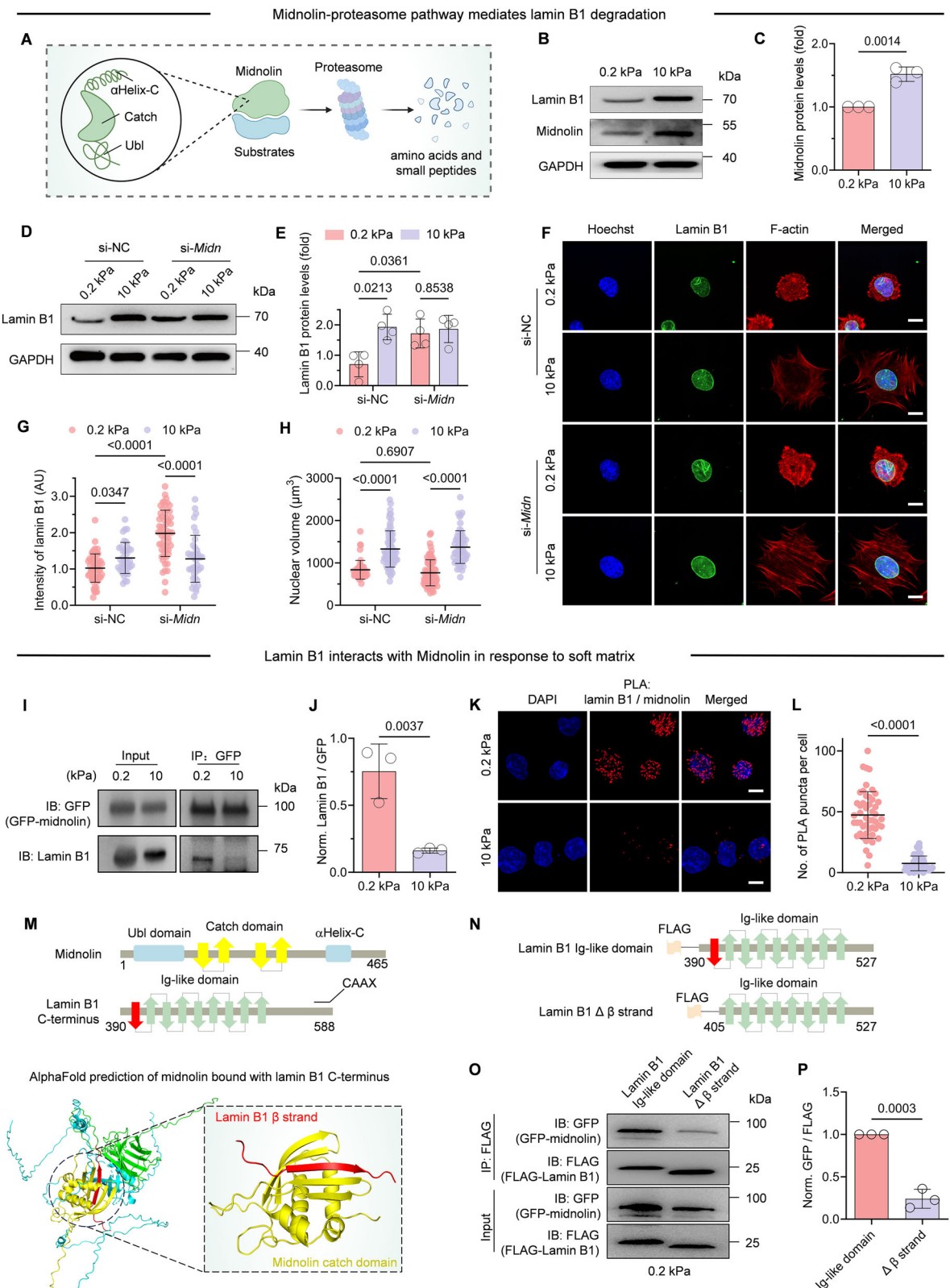

**Midnolin-proteasome pathway mediates lamin B1 degradation**

**A**

Midnolin
αHelix-C
Catch
Ubl
Substrates

Proteasome

amino acids and small peptides

**B**

| | 0.2 kPa | 10 kPa | kDa |
|---|---|---|---|
| Lamin B1 | | | 70 |
| Midnolin | | | 55 |
| GAPDH | | | 40 |

**C** Midnolin protein levels (fold) — 0.0014

**D**

| | si-NC | si-Midn | kDa |
|---|---|---|---|
| | 0.2 kPa  10 kPa | 0.2 kPa  10 kPa | |
| Lamin B1 | | | 70 |
| GAPDH | | | 40 |

**E** Lamin B1 protein levels (fold) — 0.2 kPa / 10 kPa — 0.0361, 0.0213, 0.8538

**F** Hoechst | Lamin B1 | F-actin | Merged — si-NC (0.2 kPa, 10 kPa), si-Midn (0.2 kPa, 10 kPa)

**G** Intensity of lamin B1 (AU) — 0.2 kPa / 10 kPa — <0.0001, 0.0347, <0.0001

**H** Nuclear volume (µm³) — 0.2 kPa / 10 kPa — 0.6907, <0.0001, <0.0001

**Lamin B1 interacts with Midnolin in response to soft matrix**

**I**

| | Input | IP: GFP | kDa |
|---|---|---|---|
| (kPa) | 0.2  10 | 0.2  10 | |
| IB: GFP (GFP-midnolin) | | | 100 |
| IB: Lamin B1 | | | 75 |

**J** Norm. Lamin B1 / GFP — 0.0037 — 0.2 kPa, 10 kPa

**K** PLA: DAPI | lamin B1 / midnolin | Merged — 0.2 kPa, 10 kPa

**L** No. of PLA puncta per cell — <0.0001 — 0.2 kPa, 10 kPa

**M**

Midnolin — Ubl domain, Catch domain, αHelix-C — 1 ... 465

Lamin B1 C-terminus — Ig-like domain — 390 ... 588 — CAAX

AlphaFold prediction of midnolin bound with lamin B1 C-terminus

Lamin B1 β strand
Midnolin catch domain

**N**

Lamin B1 Ig-like domain — FLAG — Ig-like domain — 390 ... 527

Lamin B1 Δ β strand — FLAG — Ig-like domain — 405 ... 527

**O**

| | Lamin B1 Ig-like domain | Lamin B1 Δ β strand | kDa |
|---|---|---|---|
| IP: FLAG | IB: GFP (GFP-midnolin) | | 100 |
| | IB: FLAG (FLAG-Lamin B1) | | 25 |
| Input | IB: GFP (GFP-midnolin) | | 100 |
| | IB: FLAG (FLAG-Lamin B1) | | 25 |
| | 0.2 kPa | | |

**P** Norm. GFP / FLAG — 0.0003 — Ig-like domain, Δ β strand

To further investigated the potential interaction between midnolin and lamin B1, we transiently transfected GFP-midnolin into human embryonic kidney (HEK)-293T cells, and seeded them onto different stiffness matrix. Consistent with observations in

C2C12 cells, lamin B1 levels significantly reduced on soft matrix in HEK293T cells (Fig. EV4C,D), suggesting conservation of this mechanoadaptive response across cell types. To study their direct binding, we treated GFP-midnolin overexpressed HEK293T cells

◀  **Figure 4.  Midnolin-proteasome pathway mediates lamin B1 degradation.**

(A) Schematic diagram of midnolin-proteasome pathway: Midnolin contains three main structural domains: the Catch domain, responsible for substrate capture; the C-terminal αHelix-C, binds to the proteasome; and the N-terminal Ubl (ubiquitin-like domain) facilitates substrate degradation. (B, C) Representative western blots and quantification analysis show the protein level changes of midnolin in C2C12 seeding onto FN and gelatin-coated PAA matrices for 0.5 h. $n = 3$ biological replicates. (D, E) Representative western blots and quantification data show the recovery of lamin B1 in response to midnolin knockdown by si-*Midn* in C2C12 after seeding cells on FN-coated PAA matrices for 0.5 h. $n = 5$ biological replicates. (F) Representative images from negative control scramble siRNA (si-NC) and midnolin knockdown (si-*Midn*) C2C12 cells seeding on 0.2 kPa and 10 kPa FN-coated PAA matrices for 0.5 h. Scale bars, 10 μm. (G, H) Quantification data of lamin B1 intensity (G) and nuclear volume (H) for fluorescence images in (F) (>50 cells for each condition). In (G), $P$ values are 0.0347, $1 \times 10^{-10}$, and $1.44 \times 10^{-8}$, respectively. In (H), $P$ values are $1 \times 10^{-10}$, 0.6907, and $5 \times 10^{-10}$, respectively. (I) GFP-midnolin HEK293T cells were pre-treated with MG132 for 3 h and then plated onto FN-coated PAA matrices for 3 h, Co-IP analysis to detect the interaction midnolin and lamin B1 was performed. IP and IB both with GFP antibody and lamin B1 antibody. (J) Normalized Lamin B1/GFP intensity in (I). $n = 3$ biological replicates. (K) In-cell lamin B1 and GFP-midnolin interactions as demonstrated by PLA in the nucleus of HEK293T cells with MG132 treatment on FN-coated PAA matrices. PLA was performed on HEK293T cells transfected with GFP-midnolin. Each red dot indicates the protein-protein interaction between GFP-midnolin and endogenous lamin B1. DAPI was used as the nuclear stain. Scale bar: 10 μm. (L) Quantification of PLA signals from experiments as in (K) (More than 50 cells for each condition. $P = 1 \times 10^{-10}$). (M) Schematic representation of midnolin and lamin B1. AlphaFold structure prediction of midnolin bound to its substrate lamin B1 (Ig-like domain) reveals an adopted β-strand capture model. (N) Schematics of lamin B1 Ig-like domain and β-strand truncation. (O) HEK293T cells were transfected with Lamin B1 Ig-like domain or β-strand truncation and cultured on FN-coated 0.2 kPa gels for 3 h. (P) Normalized midnolin/lamin B1 intensity in (O). $n = 3$ biological replicates. Data information: in (C, E, G, H, J, L, P) were presented as the mean ± SD. Two-tailed Student's *t* test was used for statistical analysis for (C, E, J, L, P). Tukey's multiple comparisons test was done for (G, H). Source data are available online for this figure.

with MG132 to stabilize potential transient interactions. Co-IP and immunoblotting assays confirmed that midnolin did interact with lamin B1 specifically on 0.2 kPa matrix but not on 10 kPa matrix (Fig. 4I,J). Additionally, in situ proximity ligation assay (PLA) revealed the specific interaction between lamin B1 and midnolin in HEK293T cells expressing GFP-midnolin on matrix, with higher PLA signals on 0.2 kPa, and the binding specificity was validated by *LMNB1* knockdown (Figs. 4K,L and EV4E–H). Therefore, both Co-IP and PLA provided the direct evidence for the interaction between midnolin and lamin B1 in those cells on 0.2 kPa matrix. Moreover, super-resolution imaging analysis further showed that the colocalization of midnolin and lamin B1 increased in C2C12 cells after MG132 treatment for 0.5 h on soft matrix but not on stiff matrix (Fig. EV4I,J).

To elucidate the structural basis of midnolin-lamin B1 interaction, we predict their binding interface through AlphaFold. Notably, the simulation revealed that midnolin's Catch domain specifically recognizes and engages a β-strand within the Ig-like domain of lamin B1, forming a five-stranded antiparallel β-sheet tertiary structure (Fig. 4M). This structural interface precisely matches the reported β-strand capture mechanism of midnolin-mediated proteasomal targeting (Gu et al, 2023). We next performed Co-IP experiments to identify β-stand of lamin B1 that are required for its interaction with midnolin. Deletion of the β-stand of lamin B1 disrupted the stable association of lamin B1 with midnolin on soft matrix (Fig. 4N–P). Notably, deletion of the C-terminal CAAX motif, a membrane-anchoring domain, did not affect lamin B1 binding to midnolin, indicating that this domain is dispensable for their stable interaction (Fig. EV4K–M). Together, these data demonstrate that the degradation of lamin B1 on soft matrix is dependent on midnolin mediated proteasomal degradation pathway.

## Loss of lamin B1 attenuates myoblast differentiation

To investigate the functional consequences of midnolin-mediated stiffness-dependent lamin B1 degradation, we first asked whether preventing lamin B1 degradation could rescue muscle atrophy on soft matrix. Knockdown of midnolin in C2C12 cells restored expression of lamin B1 and rescued myoblast differentiation on soft matrix (Fig. 5A–D). To validate the specific effect of midnolin on lamin B1 in myoblast differentiation, we reconstituted *Lmnb1*-null C2C12 cells with either wild-type (WT) lamin B1 or a mutant lacking the midnolin-binding β-strand, and then differentiated them on soft matrix. Our results showed that, the mutant group exhibited significantly higher lamin B1 protein levels on soft matrix, compared to WT group, proving that it can prevent lamin B1 from midnolin-mediated degradation (Fig. 5E–G). Most importantly, the expression of the midnolin-binding-deficient lamin B1 mutant significantly restored myoblast differentiation on soft matrix, whereas WT lamin B1 failed to do so (Fig. 5H–J), demonstrating the specific role of lamin B1 degradation in the stiffness-dependent differentiation.

We then examined the potential role of lamin B1 in myoblast differentiation. We directly used siRNA and CRISPR/Cas9 systems to verify the function of lamin B1 in myotube formation. Knockdown of *Lmnb1* reduced myoblast differentiation a lot, while knockout of *Lmnb1* completely abolished myoblast differentiation (Fig. 5K–M). This suggested that lamin B1 is required for myoblast differentiation. Subsequently, we demonstrated lamin B1 protein level indeed dynamically changed during C2C12 differentiation with a significant increase observed within the critical early stage (<3 h post-induction), implying its potential role for initiating myoblast differentiation (Fig. 5N,O). To determine its functional effect, GO analysis of upregulated genes identified non-canonical Wnt signaling as an affected pathway during this early stage (Fig. 5P). Further analysis implicated the involvement of *Wnt4*, a known regulator of skeletal muscle development (Takata et al, 2007; Tanaka et al, 2011), as a key component of these regulatory networks (Fig. 5Q). Consistent with these findings, *Lmnb1* knockdown resulted in significantly reduced *Wnt4* expression during differentiation, with the most pronounced decrease occurring at 6 h post-induction (Figs. 5R and EV5A). We next examined key myogenic regulator: *Myod1*, which drives differentiation (Olguín and Pisconti, 2012). qPCR analysis showed that *Lmnb1* knockdown caused sustained dysregulation of *Myod1* expression patterns throughout myoblast differentiation (Fig. 5S). Collectively, these results suggested that the decrease of lamin B1 protein attenuates

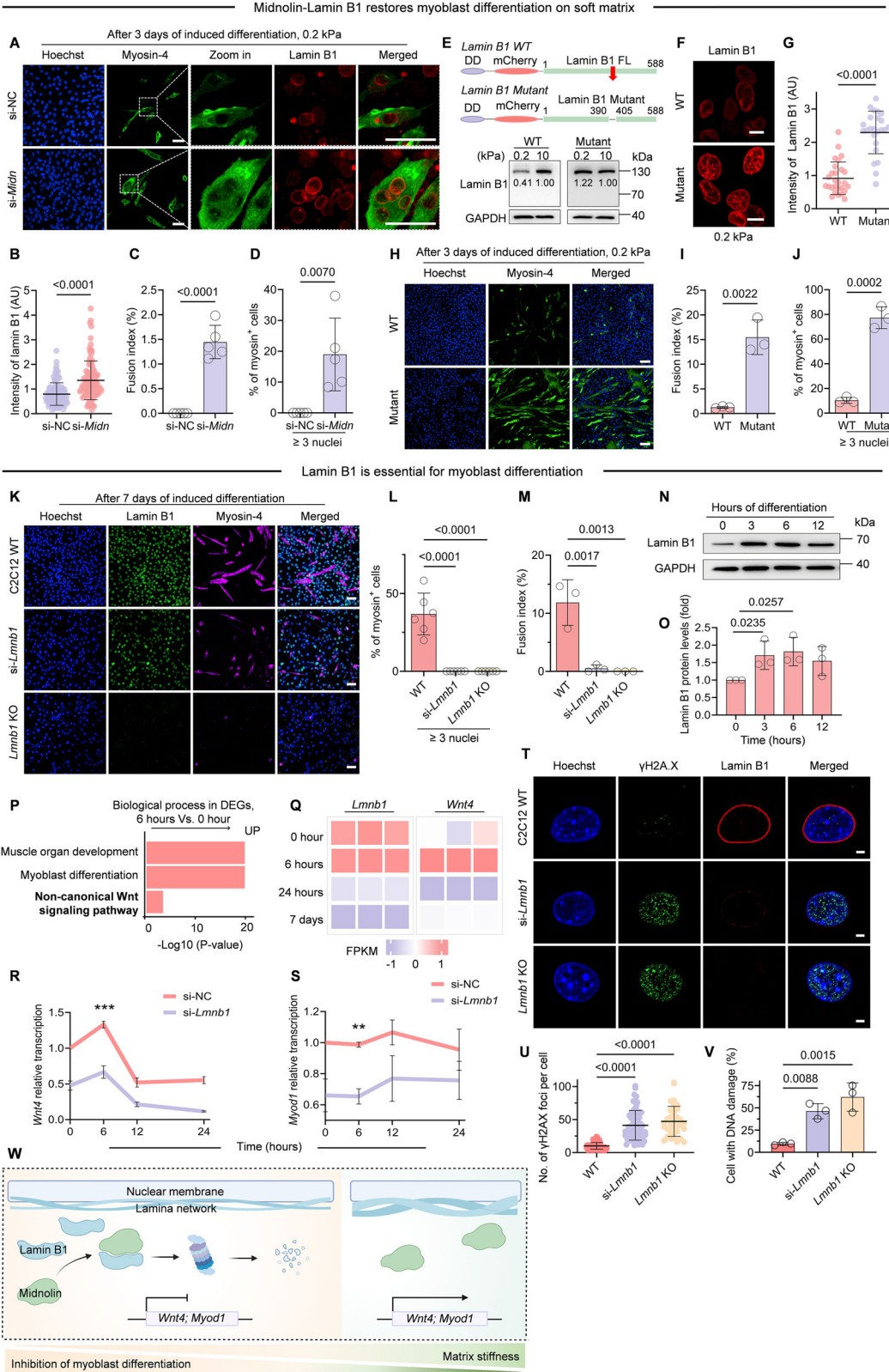

◀

**Figure 5.   Lamin B1 protein is essential for myoblast differentiation.**

(A) Representative images of si-NC and si-*Midn* C2C12 cells on FN-coated 0.2 kPa gels show the recovery of myoblast differentiation for 3 days. Scale bar, 50 μm.
(B) Quantitative analysis of fluorescence intensity of lamin B1 from (A) (>100 cells for each group, Two-tailed Student's *t* test). $P = 4.61 \times 10^{-9}$. (C, D) Quantification data of the fusion index and the percentage of myosin+ cells that ≥3 nuclei in (A). $n = 5$ biological replicates. Two-tailed Student's *t* test. $P = 1.20 \times 10^{-5}$. (E) Schematics of Inducible lamin B1 WT (full length, FL) and lamin B1 Mutant (β-strand truncation) were established by the fusion of a destabilization domain (DD). Western blot analysis of the protein levels of WT lamin B1 and mutant lamin B1 in *Lmnb1* KO C2C12 cells overexpressing either one of them on 0.2 kPa or 10 kPa gels. Cells were pre-treated with 1μM Shield1 for 12 h before seeding on FN-coated matrices. (F) Immunofluorescence of the WT lamin B1 and the mutant lamin B1 in *Lmnb1* KO C2C12 cells overexpressing either one of them with Shield1 pre-treatment (1μM) 12 h before seeding on FN-coated 0.2 kPa gels. Scale bars, 10 μm. (G) Quantitative analysis of fluorescence intensity of lamin B1 from (F) (>25 cells for each group, two-tailed Student's *t* test). $P = 1 \times 10^{-10}$. (H) Representative images of WT lamin B1 and the mutant in *Lmnb1* KO C2C12 cells on FN-coated 0.2 kPa gels showed the recovery of myoblast differentiation for 3 days. Scale bar, 100 μm. (I, J) Quantification data of the fusion index and the percentage of myosin+ cells that ≥3 nuclei in (H) ($n = 3$. Two-tailed Student's *t* test). (K) Representative images of WT, si-*Lmnb1*, and *Lmnb1* KO C2C12 cells on petri-dishes with horse serum induced differentiation for 7 days. scale bar, 100 μm. (L, M) Quantification data of the fusion index and the percentage of myosin+ cells that ≥3 nuclei in (K). One-way ANOVA/Dunnett's multiple comparisons test. In (L), $P$ values are $1.24 \times 10^{-6}$ and $1.24 \times 10^{-6}$, respectively. (N, O) Western blot analysis of lamin B1 dynamics in C2C12 during 2% horse serum induced differentiation on standard Petri-dishes, with corresponding intensity quantification data. $n = 3$ biological replicates. (P) GO terms analysis of biological processes for upregulated genes (6 h Vs. 0 h) after 2% horse serum induced differentiation. (Q) Representative genes from RNA-seq were upregulated in pink and downregulated in violet genes after 2% horse serum induced differentiation. (R, S) qPCR to examine the expression of *Wnt4* and *Myod1* during 24 h differentiation with scramble siRNA (si-NC and si-*Lmnb1*). $n = 3$ biological replicates. Two-tailed Student's *t* test. (R) $P = 9.64 \times 10^{-4}$. (S) $P = 0.0030$. (T) Representative images from WT, si-*Lmnb1*, and *Lmnb1* KO C2C12 cells on Petri-dishes after 3 days of horse serum induction. Hoechst for nuclear staining in blue, γH2AX staining in green and lamin B1 staining in red. Scale bars, 10 μm. (U, V) The Quantification of DNA damage (U) and γH2AX foci (V) in (T). $n > 40$ cells for each condition. One-way ANOVA/Dunnett's multiple comparisons test. In (U), $P$ values are $1 \times 10^{-10}$ and $1 \times 10^{-10}$, respectively. (W) Schematic model representation showing that lamin B1 protein levels are reduced by midnolin mediated proteasomal degradation on soft matrix. Consequently, this leads to reduced transcription of *Wnt4* and *Myod1*, which inhibits the process of myoblast differentiation (figure created with BioRender.com). Data information: data in (B, C, D, G, I, J, L, M, O, R, S, U, V) were presented as the mean ± SD. Source data are available online for this figure.

myoblast differentiation through downregulating differentiation-related genes in myoblasts specifically. We next examined whether lamin B1 depletion affects genomic stability during differentiation. Strikingly, *Lmnb1*-deficient C2C12 cells exhibited a significant increase in γH2AX foci (Fig. 5T–V). These findings suggest that lamin B1 maintains genomic integrity during myogenesis, and its loss leads to both DNA damage accumulation and impaired differentiation.

We further generated C2C12 cells stably overexpressing lamin B1, constitutive lamin B1 expression impaired rather than enhanced myoblast differentiation (Fig. EV5B). Time-course analysis revealed that endogenous lamin B1 protein levels progressively declined during prolonged differentiation (Fig. EV5C,D), suggesting its functional requirement is restricted to early differentiation stages. Taken together, these results demonstrate that lamin B1 mediates stiffness-dependent regulation of myoblast differentiation, where its transient early expression is required for differentiation initiation.

## Discussion

Collectively, our data discover that low matrix stiffness induces disruption of nuclear integrity in hindering myoblast differentiation. To elucidate the nuclear skeleton in regulation of nuclear integrity under mechanical stimulation, we observed lamina protein reduction in C2C12 exposed to soft matrix. Notably, we revealed that the transient lamin B1 protein decrease relays on ubiquitination-independent midnolin-proteasome degradation pathway via a β-strand capture mechanism. In addition, the soft matrix-induced lamin B1 degradation impedes myoblast differentiation into myotubes, potentially mediated by reduced expression of *Wnt4* and *Myod1* following lamin B1 knockdown. These results suggest that soft matrix enhances midnolin-lamin B1 interaction to promote lamin B1 proteasomal degradation, leading to nuclear abnormalities, DNA damage, and subsequent repression

of *Wnt4* and *Myod1* expression, ultimately impairing myoblast differentiation (Fig. 5W). These data uncover a previously unrecognized relationship whereby matrix stiffness modulates lamin B1 protein levels to govern myoblast differentiation.

Nuclear mechanics has emerged as a central focus in cellular mechanobiology research. Our findings demonstrate that ECM stiffness modulates nuclear lamina organization and transcriptional programs to drive cell fate decisions (Fig. 1), which is consistent with previous reports (De Belly et al, 2022; Nava et al, 2020). People have highlighted the ability of lamin A/C responds to mechanical stimuli at both protein level and spatial location, thereby influences the differentiation of MSCs (Buxboim et al, 2014; Ihalainen et al, 2015; Swift et al, 2013). In contrast to lamin A/C, lamin B1— while essential for nuclear membrane integrity—has been primarily characterized for its developmental functions (Chang et al, 2022; Vergnes et al, 2004), with its potential role in nuclear mechanoadaptation and cell fate regulation remaining largely unexplored. Our data show that lamin B1 decreases on soft matrix, inducing nuclear volume shrinkage and envelope wrinkling phenotypically (Figs. 1 and 2), which is similar to the phenotype in lamin B1 knockout cells reported by previous studies (Vahabikashi et al, 2022). However, the trends in lamin B1 abundance and nuclear volume are not strictly proportional. We speculate that this imperfect correlation may be attributable to compensatory contributions from lamin A/C, whose levels and organization are also regulated by ECM stiffness (Fig. 2B). This interpretation is supported by prior studies demonstrating that lamin A/C and lamin B1 differentially contribute to nuclear architecture and mechanics (Kim et al, 2017; Matias et al, 2022; Vahabikashi et al, 2022). Although how distinct lamin subtypes coordinately regulate nuclear mechanical adaptation to ECM cues remains to be elucidated.

We also found that appropriate lamin B1 protein levels are critical for early myoblast formation (Fig. 5), due to the fact that the process of early myoblast differentiation may require nuclear skeleton remodeling. Previous study showed that both LBR and lamin B2 have a tendency to be upregulated in early stage of differentiation followed by gradual decrease (Bakay et al, 2006). Another NE protein, Emerin, is required

for the perinuclear localization and inhibition of expression of *Myod1*, *Myf5*, and *Pax7* (Demmerle et al, 2013). *Myod1*, a master regulator of myogenesis, orchestrates myoblast differentiation and exhibits a two-fold upregulation during this process (Langley et al, 2002). While the regulatory relationship between lamin B1 and myogenic factors *Myod1* and *Wnt4* has remained unexplored, our study establishes for the first time that lamin B1 protein levels directly correlate with the dynamics of *Myod1* and *Wnt4* expression during myoblast differentiation. Reduction of lamin B1 level by soft matrix or direct deficiency of lamin B1 lead to DNA damage in cells during myogenic differentiation (Figs. 1 and 5). As a previous study reported, in a differentiation checkpoint where genome integrity needs to be ensured, DNA damage can impede the process of myogenic differentiation (Puri et al, 2002).

The 'half-life' of different proteins in the cell is quite variable (Correa Marrero and Barrio-Hernandez, 2021; Eldeeb et al, 2019), and this also depends on the types of the stimuli. The half-life of lamin B1 degradation induced by oncogenic injury is longer than a day via the autophagy pathway (Dou et al, 2015), whereas the ubiquitination-dependent proteasomal degradation pathway mediated by E3 ligases also requires a minimum half-life of 6 h for lamin B1 (Khanna et al, 2018; Krishnamoorthy et al, 2018). However, our results suggest that in addition to the lamin B1 degradation process described above, the response of lamin B1 to mechanical stimuli is dependent on the ubiquitination-independent proteasome pathway (Figs. 3 and 4). As latest study reported, midnolin facilitates substrate degradation with its self-contained Ubl domain, which is a much faster pathway to achieve efficient and protein degradation (Gu et al, 2023). We validated that the specificity of the interaction between midnolin and lamin B1 through the β strand truncation (Figs. 4N–P and 5E–G). Therefore, midnolin-mediated lamin B1 degradation in response to soft matrix represents a precise molecular mechanism underlying nuclear mechano-adaptation.

In summary, we have identified a novel ubiquitination-independent mechanism whereby midnolin targets lamin B1 for proteasomal degradation in response to mechanical cues, leading to nuclear integrity loss. This pathway plays a critical role in myogenesis by modulating the expression of key myogenic regulators *Myod1* and *Wnt4*. Our results establish the midnolin-mediated lamin B1 degradation pathway as a potential therapeutic lever for intervening in muscle pathologies characterized by defective nuclear mechanotransduction.

# Methods

### Reagents and tools table

| Reagent/resource | Reference or source | Identifier or catalog number |
|---|---|---|
| **Cell lines** | | |
| C2C12 (*M. musculus*) | ATCC | CRL-1772™ |
| HEK293T (*H. sapiens*) | ATCC | CRL-3216™ |
| Primary myoblasts (*M. musculus*) | Gift of Dr. Yang Zhang | N/A |
| **Recombinant DNA** | | |
| pSIN-FLAG -lamin B1 | This study | N/A |
| pSIN-FLAG -lamin B1 Δ β-strand | This study | N/A |

| Reagent/resource | Reference or source | Identifier or catalog number |
|---|---|---|
| pSIN- FLAG -lamin B1 Δ CAAX motif | This study | N/A |
| pSIN-DD-mCherry-lamin B1 WT | This study | N/A |
| pSIN- DD-mCherry-lamin B1 Mutant | This study | N/A |
| **Antibodies** | | |
| Rabbit SUN1 (1:1000) | Abcam | Cat#ab103021 |
| Rabbit SUN2 (1:5000) | Abcam | Cat#ab124916 |
| Rabbit KAP1 (1:5000) | Abcam | Cat#ab109287 |
| Rabbit Emerin (1:1000) | Cell Signaling Technology | Cat#30853 |
| Rabbit GAPDH (1:5000) | Cell Signaling Technology | Cat#2118 |
| Rabbit HP1α (1:1000) | Cell Signaling Technology | Cat#2616 |
| Anti-rabbit IgG, HRP-linked Antibody (1:10000) | Cell Signaling Technology | Cat#7074 |
| Anti-mouse IgG, HRP-linked Antibody (1:10000) | Cell Signaling Technology | Cat#7076 |
| Anti-rat IgG, HRP-linked Antibody (1:1000) | Cell Signaling Technology | Cat#7077 |
| Mouse Lamin A/C (1:1000) | Cell Signaling Technology | Cat#4777 |
| Mouse Lamin B Receptor (1:1000) | Abcam | Cat#ab232731 |
| Mouse LAP2 (1:5000) | BD Biosciences | Cat#BD611000 |
| Mouse Myosin 4 (1:1000 for WB, 1:500 for IF) | eBioscience™ | Cat#14-6503-82 |
| Mouse Lamin B1 (1:500 for IF) | Santa Cruz Biotechnology | Cat#sc-374015 |
| Rat Nup153 (1:1000) | Santa Cruz Biotechnology | Cat#sc-101544 |
| Anti-mouse IgG (H + L), F(ab')2 Fragment (Alexa Fluor® 488 Conjugate) (1:1000) | Cell Signaling Technology | Cat#4408 |
| Anti-rabbit IgG (H + L), F(ab')2 Fragment (Alexa Fluor® 594 Conjugate) (1:1000) | Cell Signaling Technology | Cat#8889 |
| Anti-rabbit IgG (H + L), F(ab')2 Fragment (Alexa Fluor® 647 Conjugate) (1:1000) | Cell Signaling Technology | Cat#4414 |
| Rabbit Lamin B1 (1:1000) | Abcam | Cat#ab16048 |
| Rabbit GFP (1:1000) | Abcam | Cat#ab290 |
| Rabbit ubiquitin (1:1000) | Cell Signaling Technology | Cat#58395S |
| Rabbit midnolin (1:200) | proteintech | Cat#18939-1-AP |
| Rabbit γH2AX (1:500) | Beyotime | Cat#C2035S |
| Mouse puromycin (1:1000) | Merckmillipore | Cat#MABE343 |
| Rabbit SUN2 (1:5000) | Proteintech | Cat#27556-1-AP |
| **Oligonucleotides** | | |
| Primers and siRNAs | This study | Dataset EV1 |
| **Chemicals, enzymes and other reagents** | | |

| Reagent/resource | Reference or source | Identifier or catalog number |
|---|---|---|
| Cycloheximide | MedChemExpress | Cat#HY-12320 |
| MG132 | MedChemExpress | Cat#HY-13259 |
| Bafilomycin A1 | MedChemExpress | Cat#HY-100558 |
| DMSO | Solarbio | Cat#D8370 |
| Shield-1 | MedChemExpress | Cat#HY-112210 |
| **Software** | | |
| GraphPad Prism 9.0.0 | https://www.graphpad.com | |
| JPK Instruments software | https://www.bruker.com | |
| ImageJ | https://imagej.nih.gov/ij/index.html | |
| Imaris | https://imaris.oxinst.com/ | |
| **Other** | | |
| Super-Resolution Microscopes (Zeiss Elyra 7) | ZEISS | |
| Dragonfly Spinning Disk Confocal Microscopy (4-laser) | Andor | |
| QuantStudio 3 Real-time PCR System | Applied Biosystems | |

## Cell culture

C2C12 (ATCC, CRL-1772™) and HEK293T (ATCC, CRL-3216™) Cells were propagated in Dulbecco's modified Eagle's medium (DMEM) (Gibco, C11995500) containing 20% Australia origin fetal bovine serum (AFBS) (Sigma, F8318) and 1% penicillin-streptomycin (Gibco, 15140122). Primary myoblasts were a kind gift of Dr. Yang Zhang, and were cultured in Nutrient Mixture F-12 Ham (Sigma, N6658) with 20% AFBS, 5 ng/mL basic FGF (YEASEN, 91330ES10) and 1% penicillin-streptomycin. Cells differentiated with DMEM containing 2% heat-inactivated horse serum (Gibco, 26050088), and cultured at 37 °C with 5% $CO_2$. All the cells were tested for mycoplasma (BeyoDirect™ Mycoplasma qPCR Detection Kit, C0303S) upon thawing of frozen stocks.

## Western blot

Western blots were implemented following standard procedures. Briefly, cells were seeded and cultured on the plates or PAA gels accordingly in 37°C incubators. After treatment with inhibitors or gels, cells were lysed using RIPA buffer. Following denaturation, lysates were loaded into 10% TGX Stain-Free polyacrylamide gels (Bio-Rad, 1610183) and transferred onto a 0.45 μm PVDF membrane (Immobilon® - P Membrane, IPVH00010). After blocking with 5% non-fat milk (CST, 9999 s), the membranes were incubated with primary antibody overnight at 4 °C and with the horseradish-peroxidase (HRP)-conjugated secondary antibody for 1 h at room temperature. ECL Western Blotting Substrate (Epizyme, SQ201) was used to detect HRP and the bands were

visualized with the ChemiDoc MP imaging system (Bio-Rad). The intensity of the bands was analyzed using ImageJ software.

## Nuclear fraction assay

In order to separate the nuclear fraction (containing lamin B1) from the cytoplasmic fraction (containing the interfering proteins e.g., BSA), nuclear and cytoplasmic proteins were isolated. Briefly, the cells were washed with DPBS twice and lysed using NE1 buffer (1 M HEPES-KOH pH 7.5, 1 M KCl, 2 M spermidine, 10% Triton X-100, and 20% glycerol and protease inhibitor). The nuclear fraction and the cytoplasmic fraction were separated by gradient centrifugation.

## Immunofluorescence

In total, 0.1 million C2C12 cells were seeded for staining experiments or 0.3 million C2C12 cells were used for myotube differentiation experiments on fibronectin (Corning, 356008) coated PAA gels or dishes. They were then fixed with 4% Paraformaldehyde (Biosharp, BL539A) for 30 min at room temperature. 0.25% Triton X-100 (Sangon Biotech, A110694) was used to permeabilize the cells for 15 min and block with 5% BSA/PBST for 1 h. Different combinations of the following primary antibodies were then used: mouse anti-Lamin B1 (1:500 Santa Cruz Biotechnology), Rabbit anti-Lamin B1 (1:1000 Abcam), and mouse anti-myosin4 (1:100 eBioscience™). Cells were incubated with primary antibodies overnight at 4 °C. They were then washed three times with TBST for 10 min. Then, the gels or dishes were incubated with secondary antibodies at room temperature for 1 h. They were then stained against Hoechst33342 (CST, 4082) or Alexa Fluor™ 568 phalloidin (ThermoFisher, A12380), and washed again with TBST. The gels were then mounted with VECTASHIELD® Antifade Mounting Medium (Vectorlabs, H-1000).

## Preparation of polyacrylamide gels and stiffness measurement

PAA gels were prepared as described previously (Tse and Engler, 2010). Briefly, Different concentrations of acrylamide and bis-acrylamide were mixed in a solution to produce gels of different rigidity. The solution also contained 10% Ammonium persulfate (APS) (Sangon Biotech, A100486), and N, N, N', N'-Tetramethy-lethylenediamine (TEMED) (Sangon Biotech, A610508). The solution was then placed on top of the glass and covered with a coverslip. After 30 min, the coverslip was removed with Dulbecco's Phosphate-Buffered Saline (DPBS). The gels were coated with 20 μg/mL fibronectin overnight at 4 °C and 0.1% gelatin (Amresco, 9764) for 1 h at 37 °C to promote cell adhesion. For subsequent experiments, only fibronectin was used for PAA gel coating to simplify the procedure. After coating, gels were washed with DPBS and then seeded cells on. The Young's modulus was measured for both FN-coated and FN&gelatin-coated PAA gels by atomic force microscopy (AFM). AFM measurements used a cylindrical tip with 1 micron end radius (SAA-SPH-1UM, Bruker AFM Probes), which was then mounted on a NanoWizard ULTRA Speed 2 system (Bruker). The stiffness was calibrated by determining a spring constant of the cantilever from the thermal fluctuations at room temperature, ranging from 0.01 to 1 N/m. The cantilever was

moved towards the stage at a rate of 2 µm/s for indentations. JPK Instruments software analyzed AFM-generated F-D curves and calculated the Young's modulus of PAA gels.

## Co-immunoprecipitation (Co-IP) assays

Co-IP was conducted as previously described (Tang et al, 2022). Briefly, HEK293T cells were transfected with GFP-midnolin using Lipofectamine 3000 reagent (Invitrogen™, L3000001). After transfection 36 h, cells were treated with MG132 for 6 h then seeded onto gels for 3 h. Cells were then washed twice with ice-cold PBS and lysed with IP lysis buffer (Thermo, 87787) containing 100×protease inhibitor (Roche, 11836170001) for 20 min on ice. After centrifugation at 4 °C, 12,000 rpm for 10 min, the protein supernatant was incubated separately with antibodies against GFP (Abcam, 290), lamin B1 (Abcam, 16048) and rabbit IgG (CST, 2729) at 4 °C overnight with rotation. The immune complexes were then incubated with Protein A/G Magnetic Beads (Vazyme, PB101) for 1 h at room temperature with rotation. For exogenous Co-immunoprecipitation (Co-IP), HEK293T cells transfected with plasmid expressing Flag-lamin B1, the supernatants mixed with Anti-Flag Nanobody-Magarose Beads (AlpaLifeBio, KTSM1361) were rotated for 2 h at 4 °C. After placing the tube on a magnet to separate the beads from the solution, the supernatant was removed, and the beads were washed three times with lysis wash buffer. Finally, 2×SDS loading buffer was added to each sample for subsequent western blot analysis.

## In vivo ubiquitination assay

C2C12 cells were seeded on gels for 30 min, and then lysed using the IP lysis buffer containing 100×protease inhibitor (Roche, 11836170001) and 5 mM/L N-ethylmaleimide (MedChemExpress, HY-D0843) to prevent de-ubiquitylation for 30 min on ice. The cell lysates were immunoprecipitated using the antibodies against lamin B1 (Abcam, 16048) and rabbit IgG, and were then subjected to immunoblotting analysis using antibody against Ub (CST, 3933S).

## Proximity ligation assay (PLA)

GFP-midnolin overexpression HEK293T Cells were fixed and permeabilized as described for IF analysis. The assays were carried out using a Duolink PLA kit (Sigma-Aldrich, DUO92002 & DUO92004) following the manufacturer's protocol. Briefly, blocking solution was added for 1 h at 37 °C. Then incubated with primary antibodies overnight at 4 °C. Each incubation step was followed by washing 2 × 5 min in Duolink® In Situ Wash Buffer. PLA probes (PLUS and MINUS) were added and incubated for 1 h at 37 °C. Ligase solution was then added and incubated for 30 min at 37 °C followed by incubation with amplification solution containing polymerase for 100 min at 37 °C and protected from light. The samples were then mounted with Duolink® In Situ Mounting Medium with DAPI (Sigma-Aldrich, DUO82040).

## Super-resolution imaging and data analysis

Cells were fixed and prepared as IF described above, and imaging was conducted using a ZEISS Elyra 7 equipped with a 100×/1.46 NA oil immersion objective. Samples were excited with 488 nm and 561 nm lasers, and structured illumination microscopy (SIM) mode was used for reconstruction. All images were processed with ZEISS ZEN software using consistent parameters, including 3D SIM reconstruction, noise filtering, and alignment, to ensure reproducibility. The Pearson's coefficients were rigorously calculated from ≥5 cells per condition using Imaris software, analyzing colocalization specifically within nuclear volumes while applying consistent thresholding and background subtraction parameters.

## Cell transfection

Plasmid DNA transfection was performed using Lipofectamine 3000 reagent (Invitrogen, L3000001), whereas siRNA transfection was performed using Lipofectamine RNAiMax (Invitrogen, 13778075). All experiments were performed 48 h after transfection.

The siRNA oligonucleotides for negative control: 5'-UUCUC CGAACGUGUCACGUUT-3' and 5'-ACGUGACACGUUCGGAG AATT-3', *Lmnb1*: 5'-AGAGUCUAGAGCAUGUUUG-3' and 5'-UUCAAGCGAAUAAACUUCCTT-3', *Midn*: 5'-GGAACAGUC CGUUAUGCAATT-3' and 5'-UUGCAUAACGGACUGUUCCT T-3'. The siRNA oligonucleotides for Human cell line: *LMNB1*: 5'-GCATTAAAGCAGCGTATC-3' (Chang et al, 2022).

## RNA isolation and qPCR

Total RNAs were extracted from cultured C2C12 cells using E.Z.N.A. Total RNA Kit I reagent (Omegabiotek, R6834-01) according to the manufacturer's instructions. Isolated RNAs were reverse-transcribed into cDNA with 5× PrimeScript RT Master Mix (Takara, RR036A). qPCR (Quantitative real-time-polymerase chain reaction) was performed with Hieff® qPCR SYBR Green Master Mix (Yeasen, 11204ES) by using the specific primer pairs. Gene expression was normalized against *GAPDH*. The qPCR primers for *GAPDH* gene were: Forward 5'-CAGAAGACTGT GGATGGCCC-3' and Reverse 5'-ATCCACGACGGACACA TTGG-3'; *Lmnb1* gene: Forward 5'- AAGGCTCTCTACGAG ACCGA-3' and Reverse 5'- TGATCTGGGCTCCACTGAGA-3'; *Myod1* gene: Forward 5'-TACAGTGGCGACTCAGATGC-3' and Reverse 5'-GTAGTAGGCGGTGTCGTAGC-3'; *Midn* gene: Forward 5'-GCGTCAACTTGCTCCCAT-3' and Reverse 5'-AACGCC TCAAAGTACCCAAG-3'. *Wnt4* gene: Forward 5'- AAGAGGAG ACGTGCGAGAAAC-3' and Reverse 5'- GTCCCTTGTGTCACC ACCTT-3'. Samples were run on a QuantStudio 3 Real-time PCR System (Applied Biosystems).

## CRISPR-mediated lamin B1 gene knockout

To knockout lamin B1 genes, we first inserted *mLmnb1* gRNA into pSpCas9(BB)-2A-GFP (PX458) (addgene#48138). Here we designed primers for two gRNAs (gRNA1: 5'-TGCAGGCGCGA-CAGGCGCGT-3', gRNA2: 5'- TCTGGAGCTTGGCGCGCTCG-3') and did cloning by using Golden Gate assembly. Plasmids were verified by Sanger sequencing. Then, 0.4 million C2C12 cells were seeded into six-well plate the day before transfection. 1.25 µg/mL *Lmnb1*-PX458-gRNA1 and 1.25 µg/mL *Lmnb1*-PX458-gRNA2 were co-transfected into C2C12 with Lipofectamine™ 3000 Reagent. 48 h post-transfection, cells were subjected to FACS to isolate GFP positive single cell clone into 96-well plates. After incubation for about a month, single clone got expanded and verified by

genotyping PCR with lamin B1-specific primers (*mLmnb1*-F: 5'-GC CTGTGGTTTGTACCTTCG-3', *mLmnb1*-R: 5'-TCATTCTTCGG GCCGTTGG-3') as well as Sanger sequencing.

## RNA-sequencing

C2C12 cells were treated and sorted as described above to harvest the total RNAs. The RNA integrity was evaluated using the Agilent 2100 Bioanalyzer. Library preparation and sequencing were performed on the Illumina NovaSeq 6000 platform. The sequencing data (Raw data) underwent quality control using fastp (v2.0) to remove adapters, trim low-quality reads (score < 20), and discard reads with an N content exceeding 10%. The processed reads were then aligned to the GRCm39 reference genome using HISAT2 software (Kim et al, 2015). Expression quantification was performed using the StringTie software (Pertea et al, 2015) against gene annotation obtained from *Mus musculus* Ensembl release 104 (Aken et al, 2016). Differential expression analysis was conducted using the DESeq2 tool (Love et al, 2014). Only genes with a fold change greater than 2 and an adjusted *P* value less than 0.05 were considered differentially regulated. The bioinformatics resources of the Database for Annotation, Visualization, and Integrated Discovery (DAVID) (Dennis et al, 2003) were used for functional annotation enrichment analysis of differentially expressed genes.

## Prediction of the interaction between lamin B1 and midnolin by AlphaFold

Residues 390–588 of *Lmnb1* sequence (UniProtKB: P14733) was paired with the MIDN sequence (UniProtKB: Q3TPJ7) as the input for multimer prediction by using AlphaFold (v2.3.2). Default reference databases and max_template_data = 2024-04-05 were used during structure prediction. We ran 10 independent predictions, and selected the top-ranking model based on the iptm+ptm score.

## Plasmid cloning

The cDNA sequences of lamin B1 were cloned into the pSIN-FLAG vector. Various truncations (Δβ-strand, residues 390 to 404; ΔCAAX motif, residues 527–588) of lamin B1 were generated from the pSIN-FLAG-lamin B1 construct to facilitate direct transfection. The cDNA sequences of midnolin were cloned into the pSIN-GFP vector. All newly created constructs in this study were rigorously verified through DNA sequencing to ensure their accuracy and integrity.

The fragment of full-length Lamin B1 was amplified from cDNA library. And then based on the PCR product of full-length Lamin B1, the DNA fragments of Lamin B1 Δ β-strand mutant was generated by multiple PCRs. And eventually pSIN-DD-mCherry-lamin B1 constructs were generated by Gibson Assembly and verified by Sanger sequencing before transfection or virus packaging.

## Construction of the DD-mCherry-Lamin B1 stable cell line

To construct the stable cell line, C2C12 *Lamin B1* KO cells were seeded onto a 24-well plate one day before infection. Cells were infected with lentivirus of pSIN-DD-mCherry-lamin B1 WT or pSIN- DD-mCherry-lamin B1 Mutant as well as 10 μg/ml

polybrene next day. After 72 h, cells with appropriate expression level of mCherry were selected using FACS.

## Puromycin incorporation assay

Puromycin incorporation assay was performed as described (Zhang et al, 2018). C2C12 cells were treated with 91 μM puromycin for 5 min after plating onto gels 25 min. Whole cell lysates were analyzed by Western Blot using a puromycin antibody (Millipore).

## Statistical analysis

All data were obtained from at least three independent experiments unless indicated otherwise. GraphPad Prism 9.0.0 (GraphPad Software) was used for statistical analysis. To calculate Pearson's correlation coefficient, the images were processed by Zeiss Elyra7 SIM (Structured Illumination Microscopy) and then analyzed by Imaris (Microscopy Image Analysis Software) for 3D colocalization. The sample size, statistical significance value and error bar graphs were indicated in figure legends.

## Graphics

Graphics created and used in the synopsis, Figs. 1A,Q, 2A, 3A,J, 4A and Fig. 5W were created with BioRender.com.

## Statement on the experimental design

No randomization or blinding procedures were applied in this study. The sample size was not predetermined by a formal statistical power calculation but was chosen based on established practices in the field for similar experimental paradigms. Cells were assessed visually for health (normal cell morphology) prior to experiments.

# Data availability

The bulk RNA sequencing data produced in this study are available in the NCBI database through accession number GSE323368. All other data that support the findings are available as a source data file.

The source data of this paper are collected in the following database record: biostudies:S-SCDT-10_1038-S44319-026-00753-0.

# Peer review information

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

## Acknowledgements

This work was financially supported by the National Natural Science Foundation of China (32471370 to QP, 12372302 to JQ, 32271360 to QL), Shenzhen Medical Research Fund (B2402010 to QP), and the Open Sharing Fund for the Large Instruments and Equipments of Shenzhen Bay Laboratory. We thank the Bioimaging Core facility at Shenzhen Bay Laboratory for providing imaging support, specifically thanking engineers Mei Yu and Shixian Huang for assistance with Super-Resolution Microscopes (Zeiss Elyra 7) and Dragonfly Spinning Disk Confocal Microscopy (4-laser), and Engineer Chunyue Zhao for her support with Atomic Force Microscopy. We also acknowledge the usage of BioRender (biorender.com) for creating the figures.

## Author contributions

**Liping Guo**: Conceptualization; Data curation; Formal analysis; Validation; Investigation; Visualization; Methodology; Writing—original draft; Writing—review and editing. **Yanjing Zhao**: Data curation; Formal analysis; Visualization; Methodology; Writing—review and editing. **Zhe Zhang**: Visualization; Methodology. **Chang Sun**: Visualization; Writing—review and editing. **Yafan Xie**: Formal analysis; Visualization. **Qin Dai**: Data curation. **Yan Yan**: Resources. **Yaoqi Zhou**: Resources. **Yang Zhang**: Resources. **Quhuan Li**: Supervision; Funding acquisition; Writing—review and editing. **Juhui Qiu**: Conceptualization; Supervision; Funding acquisition; Methodology; Writing—review and editing. **Qin Peng**: Conceptualization; Supervision; Funding acquisition; Methodology; Writing—review and editing.

Source data underlying figure panels in this paper may have individual authorship assigned. Where available, figure panel/source data authorship is listed in the following database record: biostudies:S-SCDT-10_1038-S44319-026-00753-0.

## Disclosure and competing interests statement

The authors declare no competing interests.

# Expanded View Figures

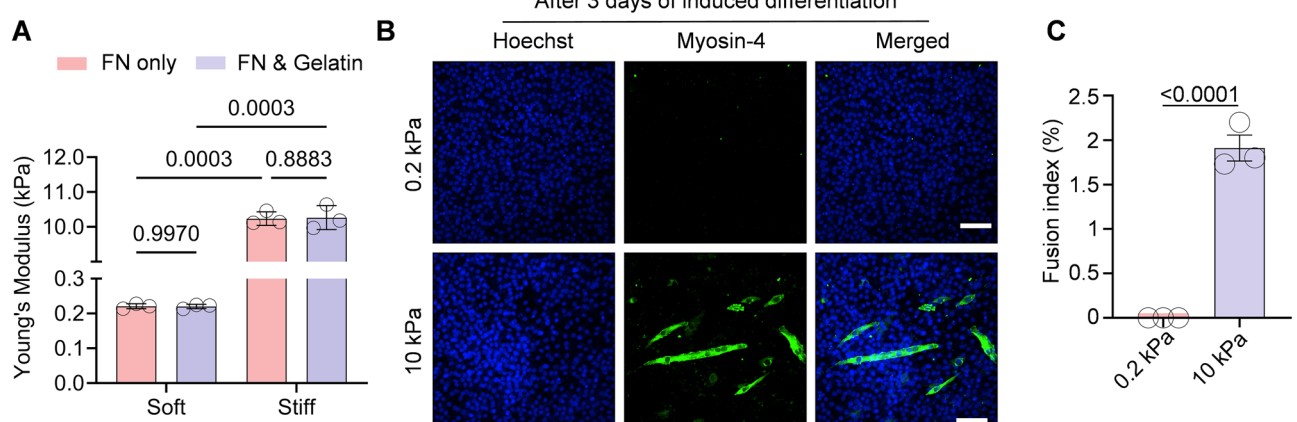

**Figure EV1. Young's modulus of PAA gels and early myoblast differentiation on different matrix.**

(A) Young's modulus (*E*, kPa) of PAA hydrogels coated with FN only versus FN combined with Gelatin (*n* = 3 biological replicates. Data are presented as the mean ± SD).

(B) Representative images of differentiated C2C12 cells on FN and gelatin-coated PAA matrices after 3 days of 2% horse serum induction. scale bar, 100 μm.

(C) Quantification data of the fusion index in (B). *n* = 3 biological replicates, Data are presented as the mean ± SD. Two-tailed Student's *t* test. *P* = 9.96 × 10⁻⁵.

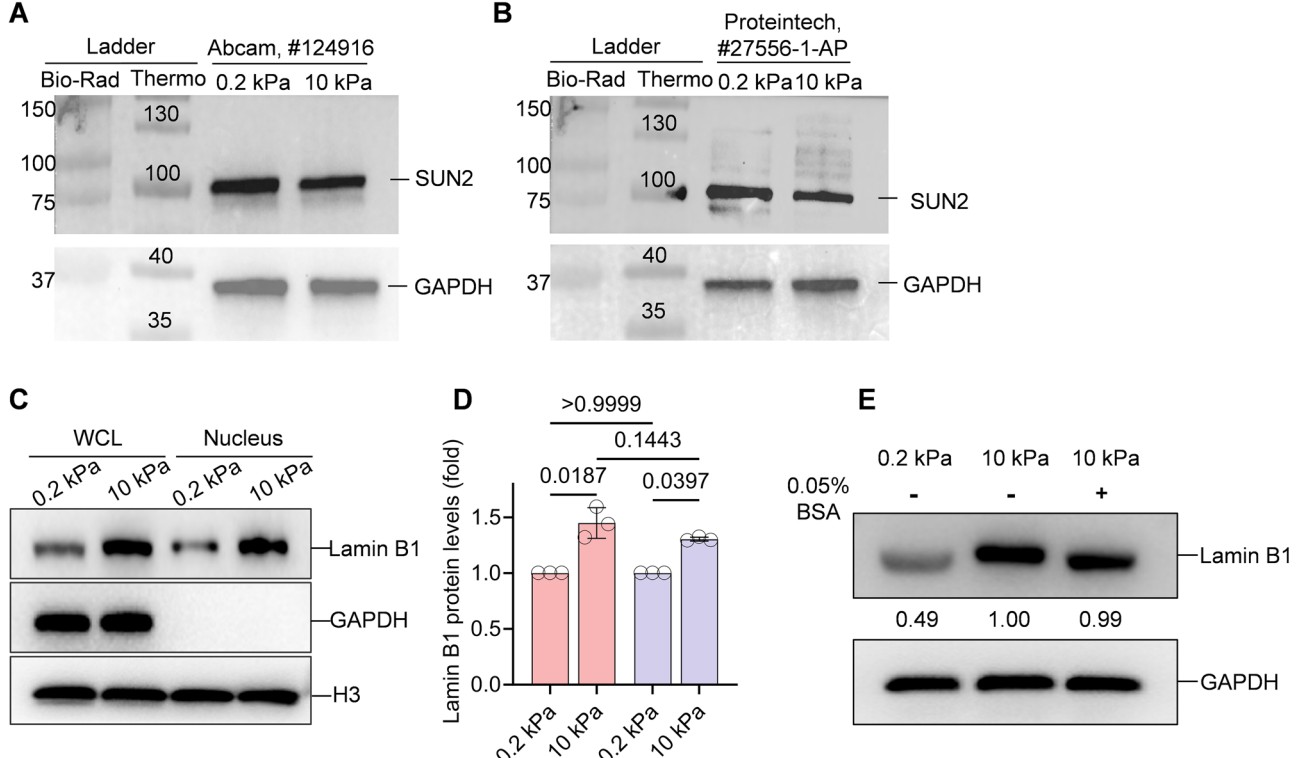

**Figure EV2. Validation of the molecular weight of SUN2 band and the band shape of Lamin B1.**

(A) SUN2 was detected by primary antibody from Abcam, which was used in this study. The band was aligned with two proteins ladder (Thermo Scientific ladder, Cat#26616 and Bio-Rad ladder, Cat# 1610374) and displayed molecular weight differently. (B) Same as (A), but detected by a different primary antibody from Proteintech. (C) Lamin B1 protein levels and band shapes were examined with whole cell lysates (WCL) or nuclear fractions (Nucleus) from FN-coated 0.2 kPa or 10 kPa gels. (D) Statistical analysis of lamin B1 protein levels among the groups from (C). $n = 3$ biological replicates. Data are presented as the mean ± SD. Tukey's multiple comparisons test. (E) Lamin B1 protein levels and band shapes were examined with or without BSA added. Protein samples were whole cell lysates from soft gels or stiff gels coated with FN.

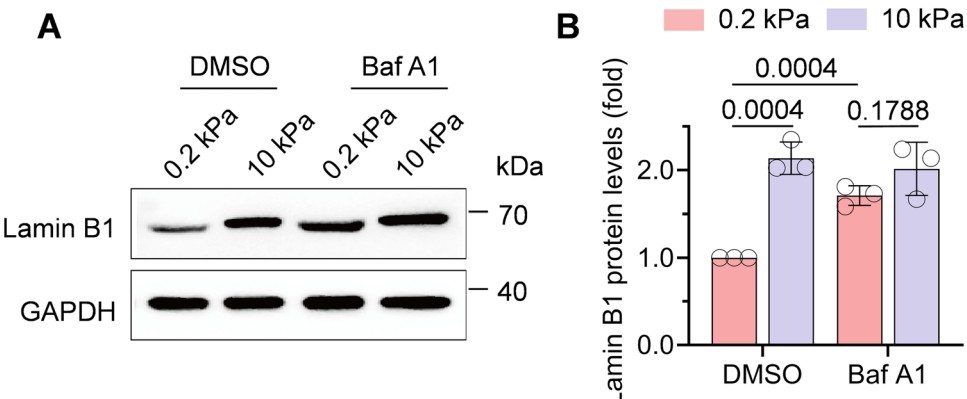

**Figure EV3.    Bafilomycin A1 treatment on matrix.**

(A, B) Western blot analysis of lamin B1 protein in C2C12 seeded onto FN and gelatin-coated PAA matrices for 30 min in the presence of degradation inhibitor (Baf A1, 100 μM). $n = 3$ biological replicates, Data are presented as the mean ± SD. Two-tailed Student's $t$ test.

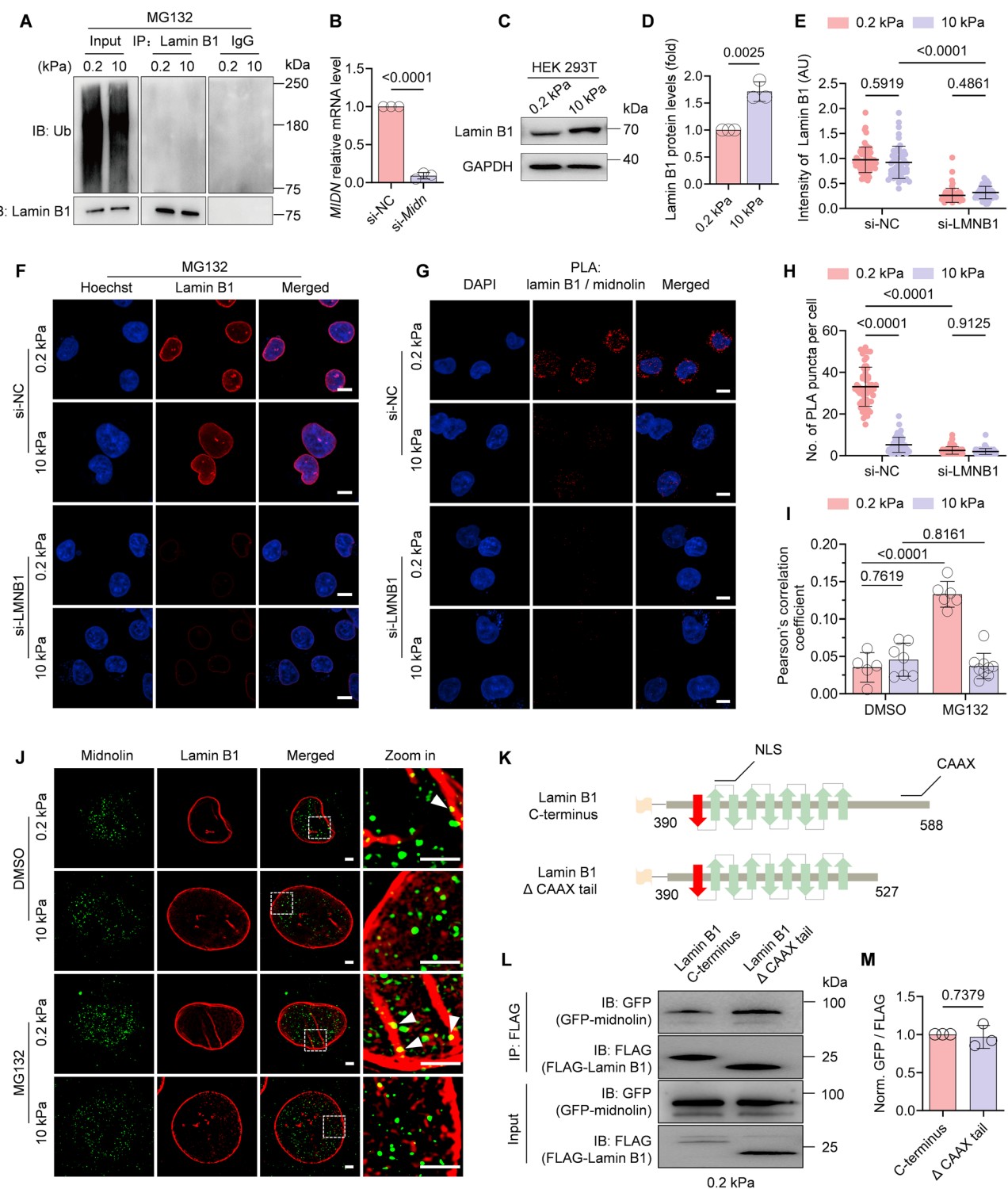

◀ **Figure EV4. Validation of the specificity of the interaction between Lamin B1 and midnolin.**

(A) The ubiquitination level of lamin B1 in C2C12 with MG132. Cells were pre-treated with MG132 for 6 h and seeded on FN and gelatin-coated PAA matrices for 0.5 h, the cells were then harvested with IP lysis buffer. IP assay was performed with anti-ubiquitin. (B) qRT-PCR to detect the expression of *Midn* transduced with si-*Midn*. $n = 3$ biological replicates. Data are presented as the mean ± SD. Two-tailed Student's *t* test was used for statistical analysis. $P = 3.04 \times 10^{-6}$. (C, D) Western blot analysis of lamin B1 proteins in HEK293T cells seeded onto FN-coated PAA matrices after 0.5 h. $n = 3$ biological replicates. (E, F) Quantification data of lamin B1 intensity and corresponding representative images from negative control scramble siRNA (si-NC) and Lamin B1 knockdown (si-*LMNB1*) HEK293T cells with MG132 treatment on FN-coated PAA matrices for 3 h. Scale bars, 10 μm. (Tukey's multiple comparisons test. Data were presented as the mean ± SD. >50 cells for each condition). In (E), *P* values are 0.5919, $2 \times 10^{-10}$, and 0.4861, respectively. (G) Representative images of PLA to detect the interaction between lamin B1 and GFP-midnolin in HEK293T cells with si-NC or si-LMNB1 on FN-coated PAA matrices for 3 h with MG132 treatment. Scale bar: 10 μm. (H) Quantification of PLA signals from experiments as in (G) (Tukey's multiple comparisons test. Data were presented as the mean ± SD. >50 cells for each condition). In (E), *P* values are $1 \times 10^{-10}$, $1 \times 10^{-10}$, and 0.9125, respectively. (I, J) Representative SIM images of midnolin (green) and lamin B1 (red) in C2C12 cells on FN and gelatin-coated PAA matrices with DMSO or MG132 treatment for 0.5 h. The yellow dots are the colocalization of midnolin and lamin B1 (white arrows). Scale bars, 2 μm. Pearson's correlation coefficient was calculated for lamin B1 and midnolin based on the imaging results in (J). $n > 5$ cells for each condition. In (I), *P* values are 0.7619, $9.26 \times 10^{-8}$, and 0.8161, respectively. (K) Schematics of lamin B1 Ig-like domain and CAAX motif truncation. (L) HEK293T cells were transfected with Lamin B1 Ig-like domain or CAAX motif truncation and cultured on FN-coated 0.2 kPa gels for 3 h. (M) Normalized midnolin/lamin B1 intensity in (J). $n = 3$ biological replicates. Data (B, D, I, M) are the mean ± SD, with two-tailed Student's *t* test.

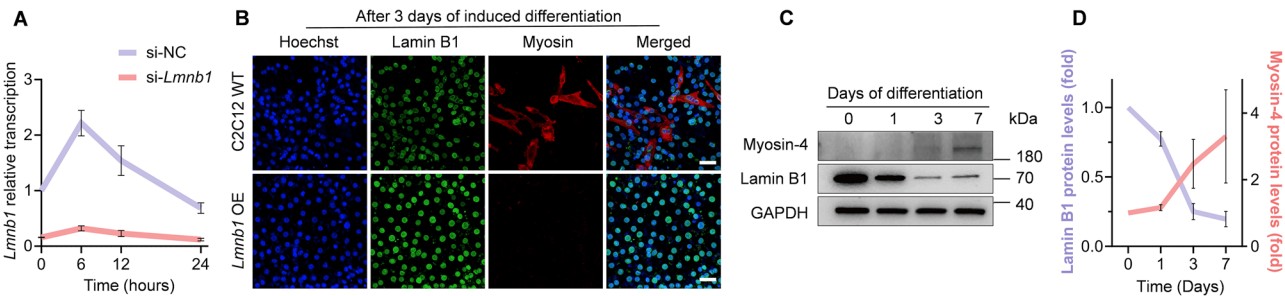

**Figure EV5. Myoblast differentiation requires moderate lamin B1 protein levels.**

(A) qPCR to examine the expression of *Lmnb1* during 24 h differentiation with scramble siRNA (si-NC) and si-*Lmnb1*. $n = 3$ biological replicates. Data are presented as the mean ± SD. (B) Representative images of WT and *Lmnb1* overexpression C2C12 cells on Petri-dishes with 2% horse serum induced differentiation for 3 days. Scale bar, 50 μm. (C, D) Western blot analysis of lamin B1 protein and Myosin-4 protein in C2C12 during 2% horse serum induced differentiation and corresponding intensity quantification data. $n = 3$ biological replicates. Data are presented as the mean ± SD.

