## [Peer Review File · EMBO Reports]

Matrix Stiffness Induces Midnolin-dependent Lamin B1 Degradation to Control Myoblast Differentiation

Liping Guo, Yanjing Zhao, Zhe Zhang, Chang Sun, Yafan Xie, Qin Dai, Yan Yan, Yaoqi Zhou, Yang Zhang, Quhuan Li, Juhui Qiu, and Qin Peng

Corresponding authors: Qin Peng (pengqin@szbl.ac.cn) , Quhuan Li (liqh@scut.edu.cn), Juhui Qiu (jhqi2008@126.com)

Review Timeline:

Submission Date:	5th Mar 24
Editorial Decision:	3rd May 24
Revision Received:	2nd Sep 25
Editorial Decision:	26th Nov 25
Revision Received:	15th Jan 26
Editorial Decision:	3rd Mar 26
Revision Received:	4th Mar 26
Accepted:	6th Mar 26

Editors: Deniz Senyilmaz Tiebe / Kurt Weir

Transaction Report:

Dear Prof. Peng,

Thank you for submitting your manuscript to EMBO Reports. We have now received three referee reports, which are included below.

I apologize for the delay in getting back to you. It took longer than anticipated to receive the full set of referee reports.

Referees express interest in the proposed mechanism by matrix stiffness regulates Lamin B1 stability and myoblast differentiation. However, they also raise concerns that preclude publication in this journal. They point out concerns regarding the methodologies used and conclusiveness of the dataset. As such, they do not recommend publication in EMBO Reports. Given such input from these recognized experts who are also experienced referees, we cannot publish your manuscript.

Thank you in any case for the opportunity to consider this manuscript. I am sorry that I cannot communicate more positive news, but nevertheless hope that you will find our referees' comments helpful.

Kind regards,

Deniz Senyilmaz Tiebe

Deniz Senyilmaz Tiebe, PhD
Editor
EMBO Reports

Referee #1:

The manuscript by Guo et al studies the differences between lamin B1 expression and degradation for cells seeded on soft versus stiff substrates, and the effects on nuclear shape. The results are interesting and provide novel insight in the stiffness-dependent regulation of lamin, which will be of interest for researchers in mechanobiology. However, there are several concerns that question the conclusions of the paper, which do not seem to be fully supported by the data. In this regard, the authors should either carry out extra experiments, or drastically tone down their conclusions. Specific comments follow.

- The main concern that I see is that none of the experiments in the paper measure the kinetics of degradation. Therefore, whether this degradation is rapid or not, or whether it is induced by exposure to soft substrates as claimed in the abstract, is not demonstrated. To show this, the authors should either use tunable stiffness gels, that can be softened with time, or at least compare conditions before/after seeding on soft/stiff substrates. What the authors actually do is to compare phenotypes on cells seeded on soft versus stiff substrates. This is of course a valid approach and leads to interesting results, but does not support that there is a "rapid degradation of lamin B1 on soft matrix" as claimed.

- In this regard, all the claims that treatments "rescue" or "restore" lamin expression should also be revised. What the authors mean is that certain treatments reduce or remove the differences between the stiff/soft condition.

- Also importantly, the specific role of the midnolin pathway is not clearly demonstrated, for different reasons:

o In fig. 2B-C, inhibiting protein translation seems to decrease lamin B1 levels on soft but not stiff substrates. The authors interpret this result by saying that "protein translation is not responsible to matrix stiffness". However, one potential alternative explanation would be that translation plays an important role in regulating lamin B1 levels on soft substrates, but not stiff substrates.

o The role of autophagy is discarded in figures EV2D-I, where the authors show that Baf A1 increases lamin B1 levels, but not nuclear volumes. However, even if there is no effect on nuclear volumes, the fact that there is an effect on lamin B1 suggests that autophagy does play a role in regulating its levels.

o In fig. 3, the authors show that Midnolin KD increases lamin 1B levels on soft substrates. However, what are the effects of this KD on stiff substrates? This is a key result that is not shown. If the effects are similar, then this simply shows the expected result that inhibiting degradation increases the amount of protein, and not a stiffness-dependent effect.

- In figure 3, the crucial IP experiments are not quantified, and there is no information regarding the number of repeats done. This should be corrected. Why were these experiments done in HEK293T cells, unlike all other data in the paper?

- In fig. 3I, colocalization is very hard to assess, and the features pointed by the white arrows are unclear. What does that mean? In the quantification, how was the pearson's coefficient calculated? This coefficient shows an increase specifically in the MG132-treated soft condition, but why this should be the case is unclear. Shouldn't we also expect co-localization in the untreated soft condition?

- Results in figure 4 are interesting. The first part, showing that stiffness affects myoblast differentiation, was already well known. The relevant part is the role of the lamin B knock down in fig. 4F-I. Surprisingly though, this part was done on glass substrates rather than soft/stiff substrates, which would have been required to assess a potential differential role of lamin B1 on soft/stiff.

- Fig. EV2A is important, as it quantifies mRNA levels of lamin B1. The authors claim that the figure shows that there are no significant differences between soft and stiff at 0.5 h of seeding, but the figure does not show the statistical analysis or its results. Please show.

- The observed trends in nuclear volumes and lamin B1 do not always correlate. Could this be due to changes in lamin A/C, which the authors also see to change with stiffness? Whereas a full study of this is out of the scope of the manuscript, it would be relevant to mention in the discussion. In this regard, the sentence in the discussion stating that "lamin B1 confers elasticity, while lamin A lends viscosity and stiffness to nuclei" is very hard to understand. What is meant here by "elasticity" that would not contribute to stiffness?

Referee #2:

In the manuscript titled "Lamin B1 Rapid Degradation by Midnolin-Proteasome Pathway on Soft Matrix" from Liping Guo and colleagues, the authors identify a role for Lamin B1 in the regulation of muscle differentiation. In particular, they show the reduction of LaminB1 levels in soft matrix. This reduction can be ascribed to a midnolin-dependent degradation mechanism. Overall, the role of Lamin B1 as a mechano-adaptor is extremely interesting and yet to be studied. Likewise, the identification of a specific and rapid Midnolin-dependent mechanism for Lamin B1 degradation is noteworthy. However, the manuscript presents some lacks and deserves further in-depth analysis to be suitable for the publication of EMBO reports.

Below main points raised by this reviewer:

- Given the RNA-seq results the authors claim that the rapid decrease of Lamin B1 protein level on soft substrates in Figure 1E is not caused by the reduction of gene transcription. However, as show in Fig EV2A, the level of LMNB1 in soft matrix (0.2 kPa) at 12 hrs is still low. Why do the authors exclude transcriptional regulation of LMNB1 levels?
- The authors postulate a role of Midnolin in degrading LAMNB1. This is inconsistent with the higher level on Midnolin in 10 kPa substrate where Lamin B1 levels are increased compare with the 0.2 kPa. I would expect, as a result of lower level of Midnolin, a higher LAMNB1 protein levels.
- In the super-resolution imaging analysis showed in Fig 3I and J is not evident the reduction of Midnolin in soft substrate compared to the stiff gels. In order to confirm the interaction among midnolin and Lamin B1, the Proximity Ligation Assay might be a valid test.
- All the experiments to build the correlation between soft substrates, Lamin B1 protein level and muscle differentiation are made on C2C12. Although the C2C12 cell line is a useful tool for studying muscle differentiation, the data obtained should be validated in a more physiological system such as that of primary cells or directly in vivo in mice induced to regenerate. In particular, the authors use MyoD and Pax7 as markers. However, as C2C12 is a cell line, Pax7 expression is not a good marker for studying muscle differentiation. The experiments shown in Fig 4C should be performed in adult stem cells (satellites) isolated from mice.
- In Figure 4 E it is not indicated in which substrate the Lamin B1 levels were verified. These levels must be measured in both soft and stiff substrates
- The results of LMNB1 deletion via si-LMNB1 and LMNB1 KO on muscle differentiation shown in Figure 4 deserve further in-depth analysis. In fact, the absence of Lamin B1 affects the global integrity of chromatin architecture. The authors should consider this and check for instance the accumulation of DNA damage that might occur in the absence of LMNB1. This might influence muscle differentiation.

Overall, the work requires further experiments aimed at proving the role of LAMNB1 during muscle differentiation in primary cells.

Referee #3:

In this study, Guo and colleagues show that lamin B1 expression is regulated by matrix rigidity. They observed that changes in matrix rigidity impact lamin B1 degradation, which is mediated by midnolin, through a ubiquitin-independent pathway. Furthermore, they found that lamin B1 expression levels affect myoblast differentiation. How matrix rigidity regulates cell differentiation is central to many aspects of biology and has been studied extensively over the last 15 years. The authors report here some interesting observations; however, the logical connection between the distinct findings is not entirely clear. Additionally, some of the authors' results and technical shortcomings question the validity of the authors' model and conclusions. Given the disparity between what is demonstrated and what the authors conclude, this model appears premature and could be misleading in this rapidly evolving field.

•More specifically, while lamin B1 depletion recapitulates certain aspects of myoblast differentiation on a soft matrix, the authors do not demonstrate its critical role in contributing to the mechanosensing of matrix rigidity. For example, they do not show whether decreasing specifically lamin B1 degradation on soft matrix recapitulates the effects observed on stiff matrix. In addition,

considering the authors' model, in which midnolin-mediated degradation is responsible for lamin B1 rapid degradation on soft matrix, it is surprising and contradictory to this model that midnolin is increased in cells cultured on a stiff matrix (figure 3B). These results suggest that matrix rigidity may affect lamin B1 through distinct pathways and significantly contradict the authors' conclusions.

- Moreover, the authors conclude that they found "new mechanisms on nuclear mechanosensing," but how matrix rigidity regulates midnolin-mediated degradation is not described, and no mechanosensing mechanism per se is described in this study.
- It is surprising that the authors coat their polyacrylamide hydrogels with gelatin. Gelatin coating has specific mechanical properties, distinct from those of polyacrylamide hydrogels, and this coating may very likely affect the elastic modulus perceived by the cells. Did the authors control the rigidity of their matrix (for example using AFM)?
- It seems that Figure 3 panels E, G, F, H and I have not been conducted on hydrogels (soft or stiff?). If this is the case, the authors can't compare these values with those obtained on a soft/stiff matrix since the control/condition used to establish the relative values are not the same. For example: "*expression level of MyoD in LMNB1 knockdown C2C12 cells decreased significantly and Pax7 level got sustained during myogenic differentiation (Figs 4H-I), which was similar to the changes of gene expression on soft substrates (Fig 4B)*"
- Midnolin and lamin B1 localization: Little technical detail is provided regarding the method of super-resolution microscopy. Did the authors perform any controls to analyze the specificity of midnolin localization (such as midnolin depletion)?
- "*The presumable explanation is that lamin B1 confers elasticity, while lamin A lends viscosity and stiffness to nuclei*" Why do the authors suggest this? Did they measure the resulting mechanical properties of the nucleus while affecting Lamin B1 expression?

** As a service to authors, EMBO Press provides authors with the ability to transfer a manuscript that one journal cannot offer to publish to another journal, without the author having to upload the manuscript data again. To transfer your manuscript to another EMBO Press journal using this service, please click on Link Not Available

We sincerely thank the reviewers for the kind and constructive feedbacks. We have conducted substantial experiments and revised our manuscript accordingly following the reviewers' suggestions. We believe that the manuscript has improved significantly as a result, for which we are grateful to all the support and help from the editor and reviewers. Our point-to-point responses are listed as follows:

Referee #1:

The manuscript by Guo et al studies the differences between lamin B1 expression and degradation for cells seeded on soft versus stiff substrates, and the effects on nuclear shape. The results are interesting and provide novel insight in the stiffness-dependent regulation of lamin, which will be of interest for researchers in mechanobiology. However, there are several concerns that question the conclusions of the paper, which do not seem to be fully supported by the data. In this regard, the authors should either carry out extra experiments, or drastically tone down their conclusions. Specific comments follow.

Response: Thank you for the insightful suggestions. To address the raised concerns comprehensively, additional experiments have been conducted as recommended. Accordingly, the conclusions in the revised manuscript have been carefully moderated to present a more balanced perspective. Point-by-point responses to each comment are provided below.

1. The main concern that I see is that none of the experiments in the paper measure the kinetics of degradation. Therefore, whether this degradation is rapid or not, or whether it is induced by exposure to soft substrates as claimed in the abstract, is not demonstrated. To show this, the authors should either use tunable stiffness gels, that can be softened with time, or at least compare conditions before/after seeding on soft/stiff substrates. What the authors actually do is to compare phenotypes on cells seeded on soft versus stiff substrates. This is of course a valid approach and leads to interesting results, but does not support that there is a "rapid degradation of lamin B1 on soft matrix" as claimed.

Response: Thank you for this valuable suggestion. We have conducted time-course experiments to measure the kinetics of lamin B1 protein degradation following seeding on soft (0.2 kPa) versus stiff (10 kPa) substrates. Our results showed that, compared to the condition before seeding, under cycloheximide (CHX) treatment to inhibit protein translation, lamin B1 levels decreased significantly on soft gels (0.2 kPa), with reductions detectable as early as 0.5 hours post-seeding and sustained low levels at 1, 3, and 6 hours (revised Fig. 3H, I). In contrast, lamin B1 levels on stiff gels (10 kPa) showed no significant changes during the same time periods. These results demonstrate that lamin B1 degradation is specifically induced by soft substrates. The experimental results have been incorporated into the revised manuscript (Page 7, Lines 29-34).

As suggested, we have replaced the term "rapid degradation of lamin B1 on soft matrix" with "stiffness-dependent degradation of lamin B1" to more accurately reflect the mechanistic conclusion in the revised manuscript.

Figure 3. Lamin B1 degradation on soft matrix through proteasome pathway. (H, I) Western blot analysis of lamin B1 proteins stability in C2C12 before seeding (-) and post-seeding onto 0.2 kPa and 10 kPa gels under cycloheximide (CHX) treatment to inhibit protein translation. $n = 3$. At 0.5 hour, compared to the condition before seeding ($t=0$), lamin B1 protein level showed a significant decrease on 0.2 kPa gel (***, $p < 0.001$), while no significant change was observed on 10 kPa gel (ns, no significance). Data are presented as the mean \pm SEM, with two-tailed Student's t -test.

2. In this regard, all the claims that treatments "rescue" or "restore" lamin expression should also be revised. What the authors mean is that certain treatments reduce or remove the differences between the stiff/soft condition.

Response: Thank you for pointing out this. As suggested, we have revised all claims regarding "rescue" or "restore" lamin expression. For instance, our data demonstrate that MG132 treatment or midnolin knockdown normalized lamin B1 levels between soft and stiff conditions, thereby reducing substrate stiffness-dependent differences (see revised Fig.3K, L and Fig. 4D, E, revised text on page 8, lines 1-3 and page 9, lines 41-43). We appreciate this insightful suggestion, which has strengthened the robustness of our conclusions by aligning terminology with the mechanistic effects observed.

Figure 3. Lamin B1 degradation on soft matrix through proteasome pathway. (K, L) Western blot analysis of lamin B1 protein in C2C12 seeded onto 0.2 kPa and 10 kPa gels for 0.5 hours in the presence of MG132 (10 μ M). $n = 3$. Data were presented as the mean \pm SD. Two-tailed Student's t -test.

Figure 4. Midnolin-proteasome pathway mediates lamin B1 degradation. (D, E) Representative western blots and quantification data show the recovery of lamin B1 in response to midnolin knockdown by si-*Midn* in C2C12 after seeding cells on 0.2 kPa and 10 kPa gels for 0.5 hours. $n = 5$. Data were presented as the mean \pm SD. Two-tailed Student's t -test.

3. Also importantly, the specific role of the midnolin pathway is not clearly demonstrated, for different reasons:

In fig. 2B-C, inhibiting protein translation seems to decrease lamin B1 levels on soft but not stiff substrates. The authors interpret this result by saying that "protein translation is not responsible to matrix stiffness". However, one potential alternative explanation would be that translation plays an important role in regulating lamin B1 levels on soft substrates, but not stiff substrates.

Response: We sincerely thank you for raising this important alternative interpretation regarding the role of protein translation in lamin B1 regulation. To directly address whether differential translation contributes to lamin B1 levels on soft versus stiff substrates, we have performed additional puromycin incorporation assays to measure global protein translation rates under both conditions. These new data (included in revised Fig. 3D, E) demonstrate that soft substrates do not significantly alter overall protein translation efficiency compared to stiff substrates, indicating that the observed reduction in lamin B1 on soft substrates is not due to changes in translation.

To further elucidate the role of the midnolin pathway, we conducted additional experiments. Our data demonstrated that (1) lamin B1 degradation on soft substrate is mediated by the proteasome pathway (revised Fig. 3). (2) the absence of substantial lamin B1 ubiquitination on soft substrate indicates that degradation occurs through a ubiquitination-independent proteasomal pathway (revised Fig. EV2A). (3) Integrated evidence from Midnolin knockdown, biochemical assays e.g., co-IP and PLA, and AlphaFold structure prediction/validation demonstrates that lamin B1 degradation on soft substrate is dependent on midnolin-mediated proteasomal degradation pathway (revised Fig. 4).

Figure 3. Lamin B1 degradation on soft matrix through proteasome pathway.

(D, E) C2C12 were treated with 91 μ M puromycin for 5 min after seeding onto gels for 25 min and immunoblotted for puromycin. $n=3$. Data were presented as the mean \pm SD. Two-tailed Student's t -test was used for statistical analysis.

4. The role of autophagy is discarded in figures EV2D-I, where the authors show that Baf A1 increases lamin B1 levels, but not nuclear volumes. However, even if there is no effect on nuclear volumes, the fact that there is an effect on lamin B1 suggests that autophagy does play a role in regulating its levels.

Response: We sincerely appreciate your insightful comment regarding the potential role of autophagy in lamin B1 regulation. We have conducted additional experiments to further investigate this important aspect, and our new data provide more comprehensive evidence regarding autophagy's involvement: LC3 lipidation analysis (new data in revised Fig. 3M-O). We examined LC3-II conversion (a gold-standard marker of autophagic flux) and found that soft matrix did not increase LC3 lipidation compared to stiff matrix (revised Fig. 3M-O), implying autophagy might not participate in regulating lamin B1 levels. On the other side, MG-132 did recovered

both lamin B1 level and nuclear volumes significantly (Fig. 3K, L, P, R), and further evidence showed that lamin B1 degradation on soft substrate is dependent on midnolin-mediated proteasomal degradation pathway (revised Fig. 4). Therefore, we have incorporated the new data and revised manuscript as below: “To examine potential autophagic degradation, we monitored LC3 lipidation, a marker of autophagy pathway activation (Fig. 3J). Quantitative analysis revealed no significant difference in the LC3-II/LC3-I ratio between soft and stiff matrix (Fig. 3M-O). Immunofluorescence results further confirmed both lamin B1 protein level and nuclear volume were significantly recovered with MG132 treatment but not with Baf A1 treatment (Fig. 3P-R). These results demonstrate the proteasome-dominant degradation of lamin B1 protein is regulated by matrix stiffness.” (Page 8, Lines 5-11)

Figure 3. Lamin B1 degradation on soft matrix through proteasome pathway.

(J) Schematic diagram of two classical protein degradation pathways. Treatment with MG132 inhibits proteasome function by blocking its proteolytic activity, and Baf A1 (bafilomycin A1) inhibits autophagy by inhibiting the fusion between autophagosomes and lysosomes (created with BioRender.com). (K, L) Western blot analysis of lamin B1 protein in C2C12 seeded onto 0.2 kPa and 10 kPa gels for 0.5 hours in the presence of MG132 (10 μM). $n = 3$. (M to O) Western blot analysis of lamin B1 protein and LC3-II/LC3-I protein in C2C12 seeded onto 0.2 kPa and 10 kPa gels for 0.5 hours in the presence of Baf A1 (100 nM). $n = 3$. (P) Immunofluorescence of C2C12 cells on 0.2 kPa substrates for 0.5 hours with DMSO or MG132 or Baf A1 treatment. Hoechst for nuclear staining in blue and lamin B1 staining in green. Scale bars, 4 μm . (Q, R) Quantitative analysis of the fluorescence intensity of lamin B1 (Q) and nuclear volume (R) for the images from (P) (> 50 cells for each condition, One-way ANOVA/Dunnett's multiple comparisons test. Data are presented as the mean \pm SD). Data information: in (L, Q, R) as the mean \pm SD, with Two-tailed Student's t -test.

5. In fig. 3, the authors show that Midnolin KD increases lamin 1B levels on soft substrates. However, what are the effects of this KD on stiff substrates? This is a key result that is not shown. If the effects are similar, then this simply shows the expected result that inhibiting degradation increases the amount of protein, and not a stiffness-dependent effect.

Response: Thank you for the suggestion. We have conducted additional experiments to examine lamin B1 levels on both stiff and soft substrates following midnolin knockdown. Our new data demonstrate that, midnolin knockdown significantly increases lamin B1 levels on soft substrates, but no statistically significant effect on stiff substrates (revised Fig. 4D-E), confirming that midnolin-mediated degradation is stiffness-dependent. We have revised the manuscript accordingly as below: "midnolin knockdown by siRNA significantly elevated lamin B1 levels on 0.2 kPa substrates but no significant effect on 10 kPa substrates, demonstrating that midnolin-mediated lamin B1 degradation is stiffness-dependent (Fig.4D, E)." (Page 9, Lines 41-43)

Figure 4. Midnolin-proteasome pathway mediates lamin B1 degradation. (D, E) Representative western blots and quantification data show the recovery of lamin B1 in response to midnolin knockdown by si-Midn in C2C12 after seeding cells on 0.2 kPa and 10 kPa gels for 0.5 hours. n = 5.

6. In figure 3, the crucial IP experiments are not quantified, and there is no information regarding the number of repeats done. This should be corrected. Why were these experiments done in HEK293T cells, unlike all other data in the paper?

Response: We sincerely appreciate your careful attention to the rigor of our Co-IP experiments and the query regarding cell type selection. To address the concerns about quantification and reproducibility, we have completed the co-IP experiments with three independent biological replicates, and presented the quantitative results in revised Figures 4I and 4J. Our data showed that, there is much more interaction between lamin B1 and midnolin on 0.2 kPa substrate than that on 10 kPa substrate. New results have been included into the revised manuscript at Lines 19-22 on Page 11.

Regarding the use of HEK293T cells for IP experiments, this choice was driven by technical necessity and biological validation. In our initial attempts, we found that endogenous midnolin antibodies exhibited suboptimal immunoprecipitation efficiency in C2C12 myoblasts, resulting in inconsistent signals and weak detection of protein interactions. Then we enhanced reliability with engineered constructs. Epitope-tagged midnolin and lamin B1 constructs demonstrated significantly higher transfection efficiency and reproducible IP results in HEK293T cells, a well-established model for protein interaction studies due to robust protein expression and ease of transfection. Critically, we confirmed that HEK293T cells recapitulate the core mechanobiological response observed in C2C12 cells: soft substrate induced a significant reduction in lamin B1 levels (revised Fig. EV4A, B). This validates HEK293T as a functionally relevant model for investigating midnolin-lamin B1 interactions under mechanical regulation.

Figure 4. Midnolin-proteasome pathway mediates lamin B1 degradation. (I) HEK293T cells were transfected with GFP-midnolin and cultured on gels for 3 hrs, Co-IP analysis to detect the interaction midnolin and lamin B1 was performed. IP and IB both with GFP antibody and lamin B1 antibody. (J) Normalized Lamin B1/GFP intensity in (I). $n = 3$. Data presented as the mean \pm SD. Two-tailed Student's *t*-test was used for statistical analysis.

Figure EV4. Lamin B1 interacts with midnolin independent on C terminal tail on 0.2 kPa. (A, B) Western blot analysis of lamin B1 proteins in HEK293T cells seeded onto 0.2 kPa and 10 kPa gels after 0.5 hours. $n = 3$. Data are the mean \pm SD, with Two-tailed Student's *t*-test.

7. In fig. 3I, colocalization is very hard to assess, and the features pointed by the white arrows are unclear. What does that mean? In the quantification, how was the pearson's coefficient calculated? This coefficient shows an increase specifically in the MG132-treated soft condition, but why this should be the case is unclear. Shouldn't we also expect co-localization in the untreated soft condition?

Response: Thank you for pointing out this. We have increased the image contrast to clearly show the yellow dots inside nucleus, which specifically indicate the colocalized midnolin and lamin B1 pointed by the white arrows. The Pearson's coefficients were rigorously calculated from ≥ 5 cells per condition using Imaris software, analyzing colocalization specifically within nuclear volumes while applying consistent thresholding and background subtraction parameters. Midnolin is known to target substrates for proteasomal degradation (Gu X et al. Science, 2023, PMID: 37616343). In untreated cells, midnolin and lamin B1 interact transiently before being rapidly degraded, making their colocalization difficult to detect. MG132 treatment stabilizes these complexes, allowing clearer visualization (as white arrows indicated). Without MG132, midnolin and lamin B1 complexes are rapidly turned over, reducing detectable colocalization. We have included the detailed information into the figure legend of Fig. EV4C and Methods in the revised manuscript at Line 29-36 on Page 19.

8. Results in figure 4 are interesting. The first part, showing that stiffness affects myoblast differentiation, was already well known. The relevant part is the role of the lamin B knock down in fig. 4F-I. Surprisingly though, this part was done on glass substrates rather than soft/stiff substrates, which would have been required to assess a potential differential role of lamin B1 on soft/stiff.

Response: Thank you. What we discovered through soft/stiff gels is that, low lamin B1 level attenuates myoblast differentiation. In original Fig 4 (Fig. 5D-F in the revised manuscript), our focus is to assess the potential role of lamin B1 in regulating myoblast differentiation. First, we conducted experiments in regular petri-dishes to reduce lamin B1 levels by siRNA or CRISPR/Cas9 KO and recapitulated the impaired differentiation phenotype observed on soft substrates (revised Fig. 5D-F). This confirms that lamin B1 depletion itself directly attenuates differentiation. Then, we demonstrated that lamin B1 levels increased significantly within the first 6 hours of differentiation initiation, accompanied with upregulation of myogenic markers *Myod1* and *Wnt4* (revised Fig. 5G-J). Depleting lamin B1 not only suppressed *Myod1* and *Wnt4* expression in early differentiation (revised Fig. 5K, L) but also induced DNA damage and genome instability (revised Fig. 5M-O), indicating its essential role in maintaining genomic integrity during myogenic commitment. These new results have been incorporated into the revised manuscript on Page 12-13.

9. Fig. EV2A is important, as it quantifies mRNA levels of lamin B1. The authors claim that the figure shows that there are no significant differences between soft and stiff at 0.5 h of seeding, but the figure does not show the statistical analysis or its results. Please show.

Response: Thank you for pointing out this. We have done statistical analysis for fig. EV2A. The statistical results showed that there are no significant differences of lamin B1 mRNA levels between soft and stiff substrates at 0.5 h post seeding. The revised heatmap and bar graph with statistics have been incorporated into revised Fig. 3B, C.

Figure 3. Lamin B1 degradation on soft matrix through proteasome pathway. (B) Representative gene from RNA-seq were upregulated in pink and downregulated in violet after 0.5 hours and 12 hours culturing on gels (0.2 kPa Vs. 10 kPa). (C) The expression of *LmnB1* from RNA-seq after 0.5 hours and 12 hours culturing on gels (0.2 kPa Vs. 10 kPa).

10. The observed trends in nuclear volumes and lamin B1 do not always correlate. Could this be due to changes in lamin A/C, which the authors also see to change with stiffness? Whereas a full study of this is out of the scope of the manuscript, it would be relevant to mention in the discussion. In this regard, the sentence in the discussion stating that "lamin B1 confers elasticity, while lamin A lends viscosity and stiffness to nuclei" is very hard to understand. What is meant here by "elasticity" that would not contribute to stiffness?

Response: Thank you for raising this important point. We agree that stiffness-induced changes in both lamin B1 and lamin A/C could collectively influence nuclear volume. Our data indicate that while lamin B1 reduction generally correlates with decreased nuclear volume on soft substrates, the imperfect correlation likely reflects compensatory mechanisms involving lamin A/C, which we also observed to be

stiffness-regulated (Fig. 2B). This interpretation aligns with established literature demonstrating that lamin A/C and lamin B1 differentially regulate nuclear morphology (Kim et al. Nat Commun, 2017, PMID: 29242553; Matias et al. Aging Cell, 2022, PMID: 34894056; Vahabikashi et al. Proc Natl Acad Sci U S A, 2022, PMID: 35439057). We fully agree that understanding how lamina networks collectively regulate nuclear adaptation to mechanical cues represents an important direction for future research. Although beyond the scope of the current study, we have revised and expanded the Discussion section on lamina composition and their corresponding functions as below: “Our data suggest that lamin B1 decreases on soft matrix, inducing nuclear volume shrinkage and envelope wrinkling phenotypically (Fig. 1-2), which is similar to the phenotype in lamin B1 knockout cells reported by previous studies (Vahabikashi et al., 2022). However, the trends in lamin B1 abundance and nuclear volume are not strictly proportional. We speculate that this imperfect correlation may be attributable to compensatory contributions from lamin A/C, whose levels and organization are also regulated by ECM stiffness (Fig. 2B). This interpretation is supported by prior studies demonstrating that lamin A/C and lamin B1 differentially contribute to nuclear architecture and mechanics (Kim, Louhghalam et al., 2017, Matias, Diniz et al., 2022, Vahabikashi et al., 2022). Although how distinct lamin subtypes coordinately regulate nuclear mechanical adaptation to ECM cues remains to be elucidated.” (Page 14, line 28-30, and Page 15, line 1-7). The revised discussion should provide clearer mechanistic interpretation while acknowledging the complexity of nuclear mechanoadaptation.

Referee #2:

In the manuscript titled "Lamin B1 Rapid Degradation by Midnolin-Proteasome Pathway on Soft Matrix" from Liping Guo and colleagues, the authors identify a role for Lamin B1 in the regulation of muscle differentiation. In particular, they show the reduction of LaminB1 levels in soft matrix. This reduction can be ascribed to a midnolin-dependent degradation mechanism.

Overall, the role of Lamin B1 as a mechano-adaptor is extremely interesting and yet to be studied. Likewise, the identification of a specific and rapid Midnolin-dependent mechanism for Lamin B1 degradation is noteworthy. However, the manuscript presents some lacks and deserves further in-depth analysis to be suitable for the publication of EMBO reports.

Below main points raised by this reviewer:

1. Given the RNA-seq results the authors claim that the rapid decrease of Lamin B1 protein level on soft substrates in Figure 1E is not caused by the reduction of gene transcription. However, as show in Fig EV2A, the level of LMNB1 in soft matrix (0.2 kPa) at 12 hrs is still low. Why do the authors exclude transcriptional regulation of LMNB1 levels?

Response: We thank you for raising this important point regarding transcriptional regulation. Our data showed that the rapid decrease in lamin B1 protein levels on soft substrates occurs at 0.5 hours (original Fig. 1E is revised Fig. 2B), but there are no significant differences of its mRNA levels between soft and stiff substrates at 0.5 h post seeding (revised Fig. 3B, C), until the transcriptional differences emerged after 12 hours. This temporal dissociation - where protein changes precede mRNA changes - strongly suggests that the initial response is mediated through post-transcriptional mechanisms rather than transcriptional regulation. While we acknowledge that

reduced transcription may contribute to maintaining low lamin B1 levels at later timepoints (12 h), it cannot account for the immediate protein reduction observed at 0.5 h, which aligns with the known short timescale of mechanotransduction responses and is further supported by our proteasome inhibition experiments (Fig. 3). We have clarified this interpretation in the revised manuscript to better emphasize that, while transcriptional regulation may play a role in sustained lamin B1 suppression in the late response, the initial decrease should occur through degradation mechanisms. (Page 7, Lines 18-22)

2. The authors postulate a role of Midnolin in degrading LAMNB1. This is inconsistent with the higher level on Midnolin in 10 kPa substrate where Lamin B1 levels are increased compare with the 0.2 kPa. I would expect, as a result of lower level of Midnolin, a higher LAMNB1 protein levels.

Response: Thank you for the comment. Following your suggestion, we performed midnolin knockdown experiments on both soft and stiff substrates. Our results clearly demonstrate that midnolin knockdown significantly increased lamin B1 levels on soft substrate, but had no significant effect on stiff substrate (Fig. 4D-E). This demonstrates a soft substrate-exclusive regulatory mechanism between midnolin and lamin B1. This substrate-specific effect indicates that midnolin-mediated lamin B1 degradation is mechanically responsive. This aligns with reports that active midnolin triggers co-degradation of bound proteins (Gu X et al. Science, 2023, PMID: 37616343), a process predominantly activated in soft environments. Our co-IP and PLA experiments showed that, there is significantly lower interaction between lamin B1 and midnolin on stiff substrates than soft substrates (Fig. 4I-L), although they both have high expression levels. Our current study focuses specifically on elucidating the mechanosensitive degradation pathway active in soft environments. The lack of interaction between lamin B1 and midnolin on stiff substrates, while interesting, represents a distinct biological scenario that falls outside the scope of this investigation into soft substrate-induced lamin B1 degradation. Our findings establish midnolin as a regulator of lamin B1 degradation in soft matrix, which represents the primary focus and novel contribution of this work.

Figure 4. Midnolin-proteasome pathway mediates lamin B1 degradation. (D, E) Representative western blots and quantification data show the recovery of lamin B1 in response to midnolin knockdown by si-Midn in C2C12 after seeding cells on 0.2 kPa and 10 kPa gels for 0.5 hours. n = 5. (I) GFP-midnolin HEK293T cells were

pre-treated with MG132 for 3 hours and then plated onto gels for 3 hours, Co-IP analysis to detect the interaction midnolin and lamin B1 was performed. IP and IB both with GFP antibody and lamin B1 antibody. (J) Normalized Lamin B1 / GFP intensity in (I). $n = 3$. Data presented as the mean \pm SD. Two-tailed Student's *t*-test was used for statistical analysis. (K) In-cell lamin B1 and GFP-midnolin interactions as demonstrated by Proximity Ligation Assay (PLA) in the nucleus of HEK293T cells with MG132 treatment. PLA was performed on HEK293T cells transfected with GFP-midnolin. Each red dot indicates the protein-protein interaction between GFP-midnolin and endogenous lamin B1. DAPI was used as the nuclear stain. Scale bar: 10 μ m. (L) Quantification of PLA signals from experiments as in (K) (Two-tailed Student's *t*-test, Data are presented as the mean \pm SD. >50 cells for each condition).

3. In the super-resolution imaging analysis showed in Fig 3I and J is not evident the reduction of Midnolin in soft substrate compared to the stiff gels. In order to confirm the interaction among midnolin and Lamin B1, the Proximity Ligation Assay might be a valid test.

Response: Thank you for the kind suggestion. Following your advice, we conducted additional Proximity Ligation Assay (PLA) experiments to further validate the interaction between midnolin and lamin B1. To accumulate their interactions, MG132 was used in PLA experiments. The PLA results demonstrate there are much more puncta on 0.2 kPa soft substrates compared to stiff substrates, indicating enhanced proximity between midnolin and lamin B1 under soft matrix conditions. This supports our model that soft substrates promote midnolin-mediated lamin B1 degradation, consistent with the colocalization data in Fig. EV3C-D and the proteasomal degradation mechanism. To elucidate the structural basis of midnolin-lamin B1 interaction, we predict their binding interface through AlphaFold. Notably, the simulation revealed that midnolin's Catch domain specifically recognizes and engages a β -strand within the Ig-like domain of lamin B1, forming a five-stranded antiparallel β -sheet tertiary structure (Fig. 4M). This structural interface precisely matches the reported β -strand capture mechanism of midnolin-mediated proteasomal targeting (Gu et al., 2023). We next performed Co-IP experiments to identify β -stand of lamin B1 that is required for its interaction with midnolin. Deletion of the β -stand of lamin B1 disrupted the stable association of lamin B1 with midnolin on soft matrix (Fig. 4N-P). These new PLA and AlphaFold related data have been incorporated into the revised manuscript (revised Fig. 4K-P) to strengthen our conclusions (Page 11, Lines 22-38).

Figure 4. Midnolin-ubiquitin pathway mediates lamin B1 degradation. (K) In-cell lamin B1 and GFP-midnolin interactions as demonstrated by Proximity Ligation Assay (PLA) in the nucleus of HEK293T cells with MG132 treatment. PLA was performed on HEK293T cells transfected with GFP-midnolin. Each red dot indicates the protein-protein interaction between GFP-midnolin and endogenous lamin B1. DAPI was used as the nuclear stain. Scale bar: 10 μ m. (J) Quantification of PLA signals from experiments as in (K) (Two-tailed Student's *t*-test, Data are presented as the mean \pm SD. >50 cells for each condition). (M) Schematic representation of midnolin and lamin B1. AlphaFold structure prediction of midnolin bound to its substrate lamin B1 (Ig-like domain) reveals an adopted β -strand capture model. (N) Schematics of lamin B1 Ig-like domain and β -strand truncation. (O) HEK293T cells were transfected with Lamin B1 Ig-like domain or β -strand truncation and cultured on 0.2 kPa gels for 3 hours. (P) Normalized midnolin/lamin B1 intensity in (O). *n* = 3.

4. All the experiments to build the correlation between soft substrates, Lamin B1 protein level and muscle differentiation are made on C2C12. Although the C2C12 cell line is a useful tool for studying muscle differentiation, the data obtained should be validated in a more physiological system such as that of primary cells or directly in vivo in mice induced to regenerate. In particular, the authors use MyoD and Pax7 as markers. However, as C2C12 is a cell line, Pax7 expression is not a good marker for studying muscle differentiation. The experiments shown in Fig 4C should be performed in adult stem cells (satellites) isolated from mice.

Response: Thank you. As suggested, we conducted additional experiments using primary mouse myoblasts to validate our findings in a more physiologically relevant system. Our results confirm that primary myoblasts similarly exhibit reduced lamin B1 levels on soft substrates (0.2 kPa) compared to stiff substrates, mirroring the behavior observed in C2C12 cells. Furthermore, during differentiation, primary myoblasts recapitulated the lamin B1 dynamics and myogenic marker expression (myosin-4) observed in our C2C12 experiments, supporting the physiological relevance of our model. These new data have been included in the revised manuscript (revised Fig. 1 and 2).

Figure 1. Soft matrix impairs myoblast differentiation and increases nuclear abnormalities and DNA damage. (C) Representative images of differentiated mouse primary myoblast cells on 0.2 kPa and 10 kPa matrices after 7 days of 2% horse serum induction. scale bar, 100 μ m. (D, E) Quantification data of the fusion index and the percentage of myosin⁺ cells that \geq 3 nuclei in (C), respectively. n = 7 (D and E).

Figure 2. Soft matrix induces a significant reduction in lamin B1 and maintains this difference over the time consistently. (F, G) Western blot analysis of lamin B1 proteins in mouse primary myoblast cells seeded onto 0.2 kPa and 10 kPa matrices after 0.5 hours. n = 3. Two-tailed Student's t-test, Data are presented as the mean \pm SD.

5. In Figure 4 E it is not indicated in which substrate the Lamin B1 levels were verified. These levels must be measured in both soft and stiff substrates

Response: Thank you. We performed the experiment of Fig. 4E on standard petri-dishes, which has been indicated in the figure legend of revised Fig. 5G in the revised manuscript. In Fig. 1-4, we have provided enough evidences on stiffness-responsive lamin B1 levels and corresponding effects on myoblast differentiation, as well as the mechanism on how midnolin mediates lamin B1 degradation on soft substrates. In Fig. 5, our results demonstrate depletion of lamin B1 itself can impair myoblast differentiation, which is independent from substrate stiffness. Therefore, we used standard petri-dishes to test the specific hypothesis that lamin B1 dynamics are essential for differentiation, regardless of substrate mechanics.

6. The results of LMNB1 deletion via si-LMNB1 and LMNB1 KO on muscle differentiation shown in Figure 4 deserve further in-depth analysis. In fact, the absence of Lamin B1 affects the global integrity of chromatin architecture. The authors should consider this and check for instance the accumulation of DNA damage that might occur in the absence of LMNB1. This might influence muscle differentiation.

Response: Thank you for the kind suggestion. We have done the additional experiments and examined DNA damage accumulation in response to soft substrates

or lamin B1 deficiency (siRNA/KO). We observed significant increases of γ H2AX foci in both si-lamin B1 group and *lmb1* KO group, compared to the WT group. These results suggest that lamin B1 loss indeed compromises nuclear integrity, contributing to DNA damage accumulation that may influence myogenic differentiation. These new data have been incorporated into the revised manuscript (Fig. 5M-O) to provide a more comprehensive mechanistic understanding.

Figure 5. Lamin B1 protein is essential for myoblast differentiation. (M) Representative images from WT, *si-Lmb1*, and *Lmb1* KO C2C12 cells on petri-dishes after 3 days of horse serum induction. Hoechst for nuclear staining in blue, γ H2AX staining in green and lamin B1 staining in red. Scale bars, 3 μ m. (N, O) The Quantification of DNA damage (N) and γ H2AX foci (O) in (M). $n > 40$ cells for each condition. One-way ANOVA/Dunnett's multiple comparisons test.

7. Overall, the work requires further experiments aimed at proving the role of LAMNB1 during muscle differentiation in primary cells.

Response: Thank you for the suggestion. We have conducted additional experiments using primary mouse myoblasts to validate our findings in a more physiologically relevant system. Our results confirm that primary myoblasts exhibit significantly reduced lamin B1 levels on soft substrates (0.2 kPa) compared to stiff substrates (10 kPa) (revised Fig. 2F, G), mirroring the behavior observed in C2C12 cells. When differentiated on 0.2 kPa gels, primary myoblasts showed significant inhibition of myogenic differentiation-directly correlating with reduced lamin B1 levels (revised Fig. 1, 2). This demonstrates that lamin B1 is essential for myoblast differentiation in primary cells, extending the mechanistic relevance beyond cell lines. These new data have been included in the revised manuscript (revised Fig. 1 and 2).

Referee #3:

In this study, Guo and colleagues show that lamin B1 expression is regulated by matrix rigidity. They observed that changes in matrix rigidity impact lamin B1 degradation, which is mediated by midnolin, through a ubiquitin-independent pathway. Furthermore, they found that lamin B1 expression levels affect myoblast differentiation. How matrix rigidity regulates cell differentiation is central to many aspects of biology and has been studied extensively over the last 15 years. The authors report here some interesting observations; however, the logical connection between the distinct findings is not entirely clear. Additionally, some of the authors'

results and technical shortcomings question the validity of the authors' model and conclusions. Given the disparity between what is demonstrated and what the authors conclude, this model appears premature and could be misleading in this rapidly evolving field.

Response: Thank you for the comments. In this study, we mainly elucidate how soft matrix suppresses myoblast differentiation (revised Fig. 5P). By integrating multidisciplinary approaches, including RNA-seq, biochemical assays, superresolution imaging, AlphaFold structural prediction, PLA, and atomic force microscopy (AFM), we demonstrate lamin B1 loss impairs myoblasts differentiation by suppressing *Myod1* and *Wnt4* expression. Particularly, we identify that midnolin mediates lamin B1 degradation via a β -strand capture mechanism in the ubiquitination-independent proteasome pathway on soft matrix. These findings identify lamin as a critical regulator of nuclear mechanoadaptation and demonstrate its indispensable for proper myogenic differentiation.

Figure 5. Lamin B1 protein is essential for myoblast differentiation. (P) Schematic model representation showing that lamin B1 protein levels are reduced by midnolin mediated proteasomal degradation on soft matrix. Consequently, this leads to reduced transcription of *Wnt4* and *Myod1*, which inhibits the process of myoblast differentiation (figure created with BioRender.com).

To address the technical concerns and clarify the logical connections between our findings, we have substantially expanded the experimental evidence. The new data now rigorously establish the mechanistic link among matrix rigidity, midnolin-mediated lamin B1 degradation, and myoblast differentiation. The revised manuscript comprehensively incorporates these updates, with key findings structured as follows:

- 1. Mechanophenotype characterization (Figure 1):** we observed the phenomenon that soft matrix impairs myoblast differentiation in both primary cells and C2C12 cell lines, accompanied by time-dependent nuclear abnormalities, including nuclear volume changes, micronuclei formation, and DNA damage accumulation, compared to stiff matrix.
- 2. Lamin B1 as a key mechanosensitive target (Figure 2):** Systematic screening of nuclear envelope proteins reveals lamin B1 exhibits the most significant reduction on soft substrates, identifying it as the primary regulator of nuclear morphology maintenance.
- 3. Proteasomal degradation mechanism (Figures 3):** we demonstrated lamin B1

reduction on soft matrix is due to proteasome pathway-dependent degradation.

- 4. Midnolin-mediated lamin B1 degradation (Figure 4):** we elucidated that enhanced interaction between midnolin and lamin B1 on soft substrate drives the degradation of lamin B1, demonstrated by loss-of-function experiments, co-IP, PLA, AlphaFold prediction and lamin B1 truncation validation.
- 5. Functional validation of lamin B1 in myoblast differentiation (Figures 5):** Midnolin knockdown on soft matrix restores primary myoblast differentiation, and directly knockdown or knockout lamin B1 also impairs myoblast differentiation, mechanistically linking lamin B1 loss to suppressed *Myod1* and *Wnt4* expression in regulating differentiation.

1. More specifically, while lamin B1 depletion recapitulates certain aspects of myoblast differentiation on a soft matrix, the authors do not demonstrate its critical role in contributing to the mechanosensing of matrix rigidity. For example, they do not show whether decreasing specifically lamin B1 degradation on soft matrix recapitulates the effects observed on stiff matrix. In addition, considering the authors' model, in which midnolin-mediated degradation is responsible for lamin B1 rapid degradation on soft matrix, it is surprising and contradictory to this model that midnolin is increased in cells cultured on a stiff matrix (figure 3B). These results suggest that matrix rigidity may affect lamin B1 through distinct pathways and significantly contradict the authors' conclusions.

Response: Thank you. To address whether lamin B1 degradation is critical for transducing matrix stiffness signals, we have performed additional experiments where we specifically blocked lamin B1 degradation on soft substrates via midnolin knockdown. Our data showed that decreasing lamin B1 degradation through midnolin knockdown dramatically rescued myoblast differentiation on 0.2 kPa gel (Revised Fig. 5A-C), demonstrating that lamin B1 degradation is a functional effector of stiffness - dependent differentiation. To strength our model of midnolin-mediated degradation on soft substrates, (1) we performed midnolin knockdown experiments on both soft and stiff substrates. Our results clearly demonstrate that midnolin knockdown significantly increased lamin B1 levels on soft substrate, but had no significant effect on stiff substrate (Revised Fig. 4D-E). This demonstrates a soft substrate-exclusive regulatory mechanism between midnolin and lamin B1. This substrate-specific effect indicates that midnolin-mediated lamin B1 degradation is mechanically responsive. This aligns with reports that active midnolin triggers co-degradation of bound proteins (Gu X et al. Science, 2023, PMID: 37616343), a process predominantly activated in soft environments. (2) Besides, our co-IP and PLA experiments showed that, there is significantly lower interaction between lamin B1 and midnolin on stiff substrates than soft substrates (Revised Fig. 4I-L), although they both have high expression levels. Our current study focuses specifically on elucidating the mechanosensitive degradation pathway active in soft environments. The lack of interaction between lamin B1 and midnolin on stiff substrates, while interesting, represents a distinct biological scenario that falls outside the scope of this investigation into soft substrate-induced lamin B1 degradation. Overall, our findings establish midnolin as a regulator of lamin B1 degradation in soft matrix, which represents the primary focus and novel contribution of this work.

Figure 5. Lamin B1 protein is essential for myoblast differentiation. (A) Representative images of si-NC and si-*Midn* C2C12 cells on 0.2 kPa gels show the recovery of myoblast differentiation for 3 days. Scale bar, 50 μ m. (B, C) Quantification data of the fusion index and the percentage of myosin⁺ cells that \geq 3 nuclei in (M). n = 5. Two-tailed Student's *t*-test.

Figure 4. Midnolin-proteasome pathway mediates lamin B1 degradation. (D, E) Representative western blots and quantification data show the recovery of lamin B1 in response to midnolin knockdown by si-*Midn* in C2C12 after seeding cells on 0.2 kPa and 10 kPa gels for 0.5 hours. n = 5. (I) GFP-midnolin HEK293T cells were pre-treated with MG132 for 3 hours and then plated onto gels for 3 hours, Co-IP analysis to detect the interaction midnolin and lamin B1 was performed. IP and IB both with GFP antibody and lamin B1 antibody. (J) Normalized Lamin B1 / GFP intensity in (I). n = 3. Data presented as the mean \pm SD. Two-tailed Student's *t*-test was used for statistical analysis. (K) In-cell lamin B1 and GFP-midnolin interactions as demonstrated by Proximity Ligation Assay (PLA) in the nucleus of HEK293 cells with MG132 treatment. PLA was performed on HEK293T cells transfected with GFP-midnolin. Each red dot indicates the protein-protein interaction between GFP-midnolin and endogenous lamin B1. DAPI was used as the nuclear stain. Scale bar: 10 μ m. (L) Quantification of PLA signals from experiments as in (K) (Two-tailed Student's *t*-test, Data are presented as the mean \pm SD. >50 cells for each condition).

2. Moreover, the authors conclude that they found "new mechanisms on nuclear mechanosensing," but how matrix rigidity regulates midnolin-mediated degradation is not described, and no mechanosensing mechanism per se is described in this study.

Response: We apologize for the confusion. The key mechanism we elucidated in this study is that, myoblasts respond to the soft substrates through midnolin mediated stiffness-dependent lamin B1 degradation, which significantly attenuates cell differentiation either by reduction of matrix rigidity or depletion of lamin B1. We performed midnolin knockdown experiments on both soft and stiff substrates. Our results clearly demonstrate that midnolin knockdown significantly increased lamin B1 levels on soft substrate, but had no significant effect on stiff substrate (Revised Fig. 4D-E). This demonstrates a soft substrate-exclusive regulatory mechanism between midnolin and lamin B1. Our co-IP and PLA experiments showed that, there is significantly lower interaction between lamin B1 and midnolin on stiff substrates than

soft substrates (Revised Fig. 4I-L), although they both have high expression levels. This substrate-specific effect indicates that midnolin-mediated lamin B1 degradation is mechanically responsive. We further conducted additional substantial experiments and demonstrated midnolin interacts with lamin B1 and mediates ubiquitination-independent degradation of lamin B1 on soft matrix through the Catch domain of midnolin engaging a β -strand within lamin B1's Ig-like domain (Revised Fig. 5). We have incorporated all the detailed description into the revised manuscript.

3. It is surprising that the authors coat their polyacrylamide hydrogels with gelatin. Gelatin coating has specific mechanical properties, distinct from those of polyacrylamide hydrogels, and this coating may very likely affect the elastic modulus perceived by the cells. Did the authors control the rigidity of their matrix (for example using AFM)?

Response: Thank you for pointing out this. We have removed the gelatin coating entirely and instead functionalized the polyacrylamide hydrogels solely with fibronectin (FN). AFM performed on these FN-only substrates ($n = 3$ biological replicates per stiffness) confirmed that the measured Young's moduli remained at 0.19 ± 0.02 kPa and 10.3 ± 0.3 kPa, i.e. within 5 % of the target values (Fig. EV1A and Methods). We have therefore updated the Methods (Page 18, Lines 32-37) and all corresponding figure legends to reflect the FN-only protocol, ensuring that the reported cellular phenotypes are unequivocally linked to the defined hydrogel stiffness.

Figure EV1. Young's modulus of PAA gels and early myoblast differentiation on different matrix. (A) Young's modulus (E , kPa) of PAA hydrogels coated with FN ($n = 3$ biological replicates).

4. It seems that Figure 3 panels E, G, F, H and I have not been conducted on hydrogels (soft or stiff?). If this is the case, the authors can't compare these values with those obtained on a soft/stiff matrix since the control/condition used to establish the relative values are not the same. For example: "expression level of MyoD in LMNB1 knockdown C2C12 cells decreased significantly and Pax7 level got sustained during myogenic differentiation (Figs 4H-I), which was similar to the changes of gene expression on soft substrates (Fig 4B)"

Response: We apologize for the confusion. To address the raised concerns comprehensively, on one side, additional experiments have been conducted, and on the other side, the description of results and conclusions have been carefully revised. What we discovered through soft/stiff gels is that, reduced lamin B1 levels impair myoblast differentiation. In original Fig 4 (now revised Fig. 5), our focus is to assess the potential role of lamin B1 in regulating myoblast differentiation. First, we conducted experiments in regular petri-dishes to reduce lamin B1 levels by siRNA or

CRISPR/Cas9 KO and recapitulated the impaired differentiation phenotype observed on soft substrates (revised Fig. 5D-F). This confirms that lamin B1 loss directly disrupts differentiation, independent of substrate mechanics. Then, we demonstrated that lamin B1 levels increased significantly within the first 6 hours of differentiation initiation, accompanied with upregulation of myogenic markers *Myod1* and *Wnt4* (revised Fig. 5J-L). Depleting lamin B1 not only suppressed *Myod1* and *Wnt4* expression in early differentiation (revised Fig. 5K, L) but also induced DNA damage and genome instability (revised Fig. 5M-O), indicating its essential role in maintaining genomic integrity during myogenic commitment. These results have been incorporated into the revised manuscript on Page 12-13.

5. Midnolin and lamin B1 localization: Little technical detail is provided regarding the method of super-resolution microscopy. Did the authors perform any controls to analyze the specificity of midnolin localization (such as midnolin depletion)?

Response: We have added technical details of super-resolution imaging as below: “Cells were fixed and prepared as IF described above, and imaging was conducted using a ZEISS Elyra 7 equipped with a 100×/1.46 NA oil immersion objective. Samples were excited with 488 nm and 561 nm lasers, and structured illumination microscopy (SIM) mode was used for reconstruction. All images were processed with ZEISS ZEN software using consistent parameters, including 3D SIM reconstruction, noise filtering, and alignment, to ensure reproducibility. The Pearson's coefficients were rigorously calculated from ≥ 5 cells per condition using Imaris software, analyzing colocalization specifically within nuclear volumes while applying consistent thresholding and background subtraction parameters.” These details have been specified in the revised Methods section on page 19, lines 30-36.

Regarding the specificity of midnolin localization in nucleus: Consistent with previous reports (Science, 2023, PMID: 37616343; Mol Cell, 2025, PMID: 40532701), our initial confocal and super-resolution imaging revealed midnolin enrichment in the nuclear compartment, which laid the foundation for investigating its interaction with nuclear lamin B1. To further validate the specificity of midnolin localization and its interaction with lamin B1, we have supplemented our data with proximity ligation assay (PLA) experiments. The results (revised Fig. 4K-L) show significantly more PLA puncta—indicating close proximity (< 40 nm) between the two proteins—in cells on soft substrates (0.2 kPa) compared to stiff substrates (10 kPa), which aligns with our model of enhanced interaction under soft mechanical conditions.

6. "The presumable explanation is that lamin B1 confers elasticity, while lamin A lends viscosity and stiffness to nuclei" Why do the authors suggest this? Did they measure the resulting mechanical properties of the nucleus while affecting Lamin B1 expression?

Response: We apologize for the confused interpretation. With the additional substantial experiments, our study mainly identified a novel ubiquitination-independent mechanism whereby midnolin targets lamin B1 for proteasomal degradation in response to mechanical cues, leading to nuclear abnormalities, DNA damage, and subsequent repression of *Myod1* and *Wnt4* expression, ultimately impairing myoblast differentiation. Nuclear volume is measured in our study to evaluate the mechanical property changes of the nucleus in response to different stiffness and different lamin B1 levels (Fig. 1, 3, 4), which is

essential for cellular function of myoblasts. According to our current results and conclusion, we have revised this part of discussion correspondingly as below: “Our data show that lamin B1 decreases on soft matrix, inducing nuclear volume shrinkage and envelope wrinkling phenotypically (Fig. 1-2), which is similar to the phenotype in lamin B1 knockout cells reported by previous studies (Vahabikashi et al., 2022). However, the trends in lamin B1 abundance and nuclear volume are not strictly proportional. We speculate that this imperfect correlation may be attributable to compensatory contributions from lamin A/C, whose levels and organization are also regulated by ECM stiffness (Fig. 2B). This interpretation is supported by prior studies demonstrating that lamin A/C and lamin B1 differentially contribute to nuclear architecture and mechanics (Kim et al, 2017; Matias et al, 2022; Vahabikashi et al., 2022). Although how distinct lamin subtypes coordinately regulate nuclear mechanical adaptation to ECM cues remains to be elucidated.” (Page 14, Lines 28-30, and Page 15, Lines 1-7)

Dear Prof. Peng

Thank you for the submission of your revised research manuscript to our journal. We have now received the set of two referee reports that is copied below.

As you will see, the referees acknowledge that the findings are interesting and that the conclusions are overall supported by the data presented but they also raise a number of concerns and have suggestions how to further strengthen the data. For example, they request more evidence showing the specific effect of Midnolin on and direct interaction with Lamin B1.

Given these constructive comments, we would like to invite you to revise your manuscript with the understanding that the referee concerns (as detailed above and in their reports) must be fully addressed and their suggestions taken on board. In particular, the expression of a mutant form of Lamin B1 to test its degradation by Midnolin and the knockdown of either Midnolin or Lamin B1 as a control to confirm their interaction will need to be incorporated. Concerns about the consistency of substrate materials can be addressed in the text and figure legends.

Please address all referee concerns in a complete point-by-point response. Acceptance of the manuscript will depend on a positive outcome of an additional round of review. Acceptance or rejection of the manuscript will therefore depend on the completeness of your responses included in the next, final version of the manuscript.

We realize that it is difficult to revise to a specific deadline. In the interest of protecting the conceptual advance provided by the work, we recommend a revision within 3 months (February 26th). Please discuss the revision progress ahead of this time with the editor if you require more time to complete the revisions.

I am also happy to discuss the revision further via e-mail or a video call, if you wish.

=====
IMPORTANT NOTE:

We perform an initial quality control of all revised manuscripts before re-review. Your manuscript will FAIL this control and the handling will be delayed IN CASE the following APPLIES:

- 1) A data availability section providing access to data deposited in public databases is missing. If you have not deposited any data, please add a sentence to the data availability section that explains that.
- 2) Your manuscript contains statistics and error bars based on $n=2$. Please use scatter blots in these cases. No statistics should be calculated if $n=2$.

=====

- 2) individual production quality figure files as .eps, .tif, .jpg (one file per figure). Please download our Figure Preparation Guidelines (figure preparation pdf) from our Author Guidelines pages <https://www.embopress.org/page/journal/14693178/authorguide> for more info on how to prepare your figures.

4) a complete author checklist, which you can download from our author guidelines (<<https://www.embopress.org/page/journal/14693178/authorguide>>). Please insert information in the checklist that is also reflected in the manuscript. The completed author checklist will also be part of the RPF.

5) Please note that all corresponding authors are required to supply an ORCID ID for their name upon submission of a revised manuscript (<<https://orcid.org/>>). Please find instructions on how to link your ORCID ID to your account in our manuscript tracking system in our Author guidelines (<<https://www.embopress.org/page/journal/14693178/authorguide#authorshipguidelines>>)

6) We replaced Supplementary Information with Expanded View (EV) Figures and Tables that are collapsible/expandable online. A maximum of 5 EV Figures can be typeset. EV Figures should be cited as 'Figure EV1, Figure EV2' etc... in the text and their respective legends should be included in the main text after the legends of regular figures.

7) Please include a dedicated "Data Availability" section at the end of the Methods (suggested wording: "The [structural coordinates | microarray | mass spectrometry] data from this publication have been deposited to the [name of the database] database [URL] and assigned the identifier [accession | permalink | hashtag]."). Should this not apply, this should still be stated as "This study includes no data deposited in external repositories."

Additional information on source data and instruction on how to label the files are available <<https://www.embopress.org/page/journal/14693178/authorguide#sourcedata>>

10) Figure legends and data quantification:
The following points must be specified in each figure legend:

- the name of the statistical test used to generate error bars and P values,
 - the EXACT p-values,
 - the number (n) of independent experiments (please specify technical or biological replicates) underlying each data point,
 - the nature of the bars and error bars (s.d., s.e.m.)
- If the data are obtained from n {less than or equal to} 5, show the individual data points in addition to the SD or SEM.
- If the data are obtained from n {less than or equal to} 2, use scatter blots showing the individual data points.

See also the guidelines for figure legend preparation:
<https://www.embopress.org/page/journal/14693178/authorguide#figureformat>

11) Our journal encourages inclusion of *data citations in the reference list* to directly cite datasets that were re-used and obtained from public databases. Data citations in the article text are distinct from normal bibliographical citations and should directly link to the database records from which the data can be accessed. In the main text, data citations are formatted as follows: "Data ref: Smith et al, 2001" or "Data ref: NCBI Sequence Read Archive PRJNA342805, 2017". In the Reference list, data citations must be labeled with "[DATASET]". A data reference must provide the database name, accession number/identifiers and a resolvable link to the landing page from which the data can be accessed at the end of the reference. Further instructions are available at <<https://www.embopress.org/page/journal/14693178/authorguide#referencesformat>>.

12) All Materials and Methods need to be described in the main text using our 'Structured Methods' format. According to this format, the Methods section includes a Reagents and Tools Table (listing key reagents, experimental models, software and relevant equipment and including their sources and relevant identifiers) followed by a Methods and Protocols section describing the methods, ideally using a step-by-step protocol format. The aim is to facilitate adoption of the methodologies across labs. Please download and fill our Reagents and Tools Table template (.docx), which you can find in our author guidelines: <https://www.embopress.org/page/journal/14693178/authorguide#structuredmethods>.

13) As part of the EMBO publication's Transparent Editorial Process, EMBO Reports publishes online a Review Process File to accompany accepted manuscripts. This File will be published in conjunction with your paper and will include the referee reports, your point-by-point response and all pertinent correspondence relating to the manuscript.

Yours sincerely,

Kurt Weir
Editor
EMBO Reports

Referee #1:

The authors have adequately addressed my concerns and the manuscript is now ready for publication. I commend the authors on their thorough revision work.

Referee #3:

The authors have conducted several experiments to address previous comments, presenting interesting results regarding the role of Lamin B1 in myoblast differentiation and Midnolin-dependent degradation. However, a few critical initial comments have not been fully addressed, and some conclusions still require revision.

A primary concern remains regarding the specificity of Lamin B1's involvement in matrix rigidity mechanosensing. While depleting Lamin B1 affects differentiation, its specific role on soft matrices has not been directly tested. The revised manuscript relies on Midnolin depletion data to address this; however, Midnolin regulates the degradation of several transcription factors involved in the mechanoresponse (Gu et al., 2023). To demonstrate Lamin B1 specificity, the authors should prevent its degradation by expressing a mutant that cannot bind to Midnolin. Alternatively, the discussion must acknowledge that the observed effects could stem from changes in Midnolin activity or its targeting of substrates other than Lamin B1 on soft matrices. A significant concern also remains regarding the consistency of the substrate materials. The authors state they have "removed the gelatin coating entirely and instead functionalized the polyacrylamide hydrogels solely with fibronectin (FN)." This raises the question of whether all presented data-both new and previously submitted-exclusively use the FN-coated matrix. It appears unlikely that all previous experiments were repeated, suggesting the current manuscript may contain a mix of ECM methodologies. To ensure data consistency and reproducibility, the authors must explicitly confirm which coating methodology was used for each specific result.

Regarding controls for the Lamin B1/Midnolin interaction, the PLA-while interesting-cannot serve as a sufficient control, as the signal may partially result from non-specific antibody binding. The gold standard for verifying specificity is the depletion of the target protein (Midnolin) or its partner (Lamin B1) to confirm that the interaction signal is indeed reduced.

Additional minor comments: several Western blot clarifications are required: the specific LAP2 isoform displayed must be

identified: The apparent molecular weight of SUN2 (~100 kDa) appears unusual; it is typically expected to be around 80-85 kDa. Did the authors attempt to validate this with a different antibody? More importantly, the distorted shape of the Lamin B1 band only in lysate from cells cultured on soft ECM is problematic. The shape of a Western blot band can be altered due to the presence of an abundant peptide or protein with a similar molecular weight, creating an electrophoresis artifact. While the Immunofluorescence (IF) data shows similar trends, the authors should address the band shape in the results section. (Some commercial acrylamide gels are known to efficiently absorb BSA from the culture medium (particularly on soft ECMs). This BSA is released during cell lysis and can distort bands near its molecular weight (~66 kDa), which is close to Lamin B1) and the unusual apparent molecular weight of SUN2 (~100 kDa versus the expected 80-85 kDa) would benefit from additional validation with a different antibody. The authors should also address the distorted shape of the Lamin B1 band in soft ECM lysates. This distortion is likely an electrophoresis artifact caused by the hydrogels absorbing BSA from the culture medium, which is then released during lysis near the molecular weight of Lamin B1.

We sincerely thank the reviewers for the kind and constructive feedbacks. We have conducted substantial experiments and revised our manuscript accordingly following the reviewers' suggestions. We believe that the manuscript has improved significantly as a result, for which we are grateful to all the support and help from the editor and reviewers. Our point-to-point responses are listed as follows:

Referee #1:

The authors have adequately addressed my concerns and the manuscript is now ready for publication. I commend the authors on their thorough revision work.

Response: Thank you very much for the invaluable suggestions, which improved our manuscript significantly.

Referee #3:

The authors have conducted several experiments to address previous comments, presenting interesting results regarding the role of Lamin B1 in myoblast differentiation and Midnolin-dependent degradation. However, a few critical initial comments have not been fully addressed, and some conclusions still require revision.

Response: Thank you for the kind comments. To fully address the raised concerns in the initial review, additional experiments have been conducted as recommended. Accordingly, the conclusions in the revised manuscript have been carefully moderated to present a more balanced perspective. Point-by-point responses to each comment are provided below.

1. A primary concern remains regarding the specificity of Lamin B1's involvement in matrix rigidity mechanosensing. While depleting Lamin B1 affects differentiation, its specific role on soft matrices has not been directly tested. The revised manuscript relies on Midnolin depletion data to address this; however, Midnolin regulates the degradation of several transcription factors involved in the mechanoresponse (Gu et al., 2023). To demonstrate Lamin B1 specificity, the authors should prevent its degradation by expressing a mutant that cannot bind to Midnolin. Alternatively, the discussion must acknowledge that the observed effects could stem from changes in Midnolin activity or its targeting of substrates other than Lamin B1 on soft matrices.

Response: Thank you for this excellent suggestion. To directly test the specificity of Lamin B1, we have followed your advice and generated a Lamin B1 mutant lacking the β -strand, the specific domain responsible for interacting with midnolin. We expressed either the full-length (WT) Lamin B1 or the mutant in *Lmnb1* KO C2C12 cells and differentiated them on soft matrix. Our results showed that, the mutant group exhibited significantly higher lamin B1 protein levels on soft matrix, compared to WT group, proving that it can prevent lamin B1 from midnolin-mediated degradation (**Fig. 5E-G**). Most importantly, the expression of the midnolin-binding-deficient lamin B1 mutant significantly restored myoblast differentiation on soft matrix, whereas WT lamin B1 failed to do so (**Fig. 5H-J**), demonstrating the specific role of lamin B1 degradation in the stiffness-dependent differentiation. These new data provide direct genetic evidence supporting the specific role of lamin B1 degradation in the mechanoresponse on soft matrix. We believe this experiment strongly addresses the reviewer's concern and

greatly strengthens our conclusion. We have included new results and corresponding conclusion in the revised manuscript (*Page 12, Lines 8-16*).

Figure 5. Lamin B1 protein is essential for myoblast differentiation. (E) Schematics of Inducible lamin B1 WT (full length, FL) and lamin B1 Mutant (β -strand truncation) were established by the fusion of a destabilization domain (DD). Western blot analysis of the protein levels of WT lamin B1 and mutant lamin B1 in *Lmnb1* KO C2C12 cells overexpressing either one of them on 0.2kPa or 10 kPa gels. Cells were pre-treated with 1 μ M Shield1 for 12 hours before seeding on FN-coated matrices. (F) Immunofluorescence of the WT lamin B1 and the mutant lamin B1 in *Lmnb1* KO C2C12 cells overexpressing either one of them with Shield1 pre-treatment (1 μ M) 12 hours before seeding on FN-coated 0.2 kPa gels. Scale bars, 10 μ m. (G) Quantitative analysis of fluorescence intensity of lamin B1 from (F) (> 25 cells for each group, Two-tailed Student's t-test). (H) Representative images of WT lamin B1 and the mutant in *Lmnb1* KO C2C12 cells on FN-coated 0.2 kPa gels showed the recovery of myoblast differentiation for 3 days. Scale bar, 100 μ m. (I, J) Quantification data of the fusion index and the percentage of myosin⁺ cells that ≥ 3 nuclei in (H) (n = 3. Two-tailed Student's t-test).

2. A significant concern also remains regarding the consistency of the substrate materials. The authors state they have "removed the gelatin coating entirely and instead functionalized the polyacrylamide hydrogels solely with fibronectin (FN)." This raises the question of whether all presented data-both new and previously submitted-exclusively use the FN-coated matrix. It appears unlikely that all previous experiments were repeated, suggesting the current manuscript may contain a mix of ECM methodologies. To ensure data consistency and reproducibility, the authors must explicitly confirm which coating methodology was used for each specific result.

Response: Thank you for pointing out this. We completely agree that clarity on this matter is paramount. To address this concern thoroughly, (1) we have explicitly stated the ECM coating method in the respective figure legends in the revised manuscript. (2) We have also revised the Methods section and included the details of two coating methods. The Method section of "Preparation of polyacrylamide gels and stiffness

measurement” was revised as below: “*The gels were coated with 20 μg/mL fibronectin overnight at 4°C and 0.1% gelatin (Amresco, 9764) for 1 hour at 37°C to promote cell adhesion. For subsequent experiments, only fibronectin was used for PAA gel coating to simplify the procedure. After coating, gels were washed with PBST and then seeded cells on. The Young’s modulus was measured for both FN-coated and FN&gelatin-coated PAA gels by atomic force microscopy (AFM). AFM measurements used a cylindrical tip with 1 micron end radius (SAA-SPH-1UM, Bruker AFM Probes), which was then mounted on a NanoWizard ULTRA Speed 2 system (Bruker). The stiffness was calibrated by determining a spring constant of the cantilever from the thermal fluctuations at room temperature, ranging from 0.01~1 N/m. The cantilever was moved towards the stage at a rate of 2 μm/s for indentations. JPK Instruments software analyzed AFM-generated F-D curves and calculated the Young’s modulus of PAA gels.*” (Pages 18-19, Lines 38-41, 1-7). (3) we performed additional AFM measurements to compare the stiffness between polyacrylamide hydrogels coated with Fibronectin (FN) only versus FN combined with Gelatin (n = 3 biological replicates). Our results showed that, there was no statistically significant difference between the two coating conditions for either soft or stiff matrix. The measured Young’s moduli remained consistent across groups (Soft: ~0.2 kPa; Stiff: ~10.2 kPa) (**Figure EV1A**), adhering strictly to the target values, indicating the addition of gelatin does not alter the bulk stiffness of the gels. And the core cellular responses we measured (e.g., similar trends of lamin B1 protein levels in Fig. 2H and Fig. 4D) were no difference between two coating methods. These confirm that the mechanical microenvironments are identical regardless of the specific coating protocol, ensuring the comparability and reproducibility of our data.

Figure EV1. Young’s modulus of PAA gels and early myoblast differentiation on different matrix. (A) Young’s modulus (E , kPa) of PAA hydrogels coated with FN only versus FN combined with Gelatin (n= 3 biological replicates. Data are presented as the mean \pm SD).

3. Regarding controls for the Lamin B1/Midnolin interaction, the PLA-while interesting-cannot serve as a sufficient control, as the signal may partially result from non-specific antibody binding. The gold standard for verifying specificity is the depletion of the target protein (Midnolin) or its partner (Lamin B1) to confirm that the interaction signal is indeed reduced.

Response: Thank you for the suggestion. We have performed the recommended depletion experiments to provide conclusive evidence. We employed siRNA to knock down Lamin B1 expression in our system. Efficient knockdown of Lamin B1 was verified by Immunofluorescence (**Fig. EV4E, F**). The PLA signal for the Lamin B1/Midnolin interaction was essentially abolished in *LMNB1* knockdown cells (**Fig. EV4G, H**). These new results robustly demonstrate that the PLA signal is specifically dependent on the presence of both Lamin B1 and midnolin, effectively excluded false positives caused by non-specific antibody binding. These new PLA data have been incorporated into the revised manuscript (**Fig. EV4E-H**) to strengthen our conclusions

Figure EV4. Validation of the specificity of the interaction between Lamin B1 and midnolin. (E, F) Quantification data of lamin B1 intensity and corresponding representative images from negative control scramble siRNA (si-NC) and Lamin B1 knockdown (si-LMNB1) HEK293T cells with MG132 treatment on FN-coated PAA matrices for 3 hours. Scale bars, 10 μ m. (Tukey's multiple comparisons test. Data were presented as the mean \pm SD. >50 cells for each condition). (G) Representative images of PLA to detect the interaction between lamin B1 and GFP-midnolin in HEK293T cells with si-NC or si-LMNB1 on FN-coated PAA matrices for 3 hours with MG132 treatment. Scale bar: 10 μ m. (H) Quantification of PLA signals from experiments as in (G) (Tukey's multiple comparisons test. Data were presented as the mean \pm SD. >50 cells for each condition).

4. Additional minor comments: several Western blot clarifications are required: the specific LAP2 isoform displayed must be identified:

Response: Thank you. We apologize for the confusion. The specific LAP2 isoform is LAP2 β , which has been indicated in the revised manuscript, including main text, Figure 2B-C, and the corresponding figure legend, to ensure clarity.

5. The apparent molecular weight of SUN2 (~100 kDa) appears unusual; it is typically expected to be around 80-85 kDa. Did the authors attempt to validate this with a different antibody? More importantly, the distorted shape of the Lamin B1 band only in lysate from cells cultured on soft ECM is problematic. The shape of a Western blot band can be altered due to the presence of an abundant peptide or protein with a similar molecular weight, creating an electrophoresis artifact. While the Immunofluorescence (IF) data shows similar trends, the authors should address the band shape in the results section. (Some commercial acrylamide gels are known to efficiently absorb BSA from the culture medium (particularly on soft ECMs). This BSA is released during cell lysis and can distort bands near its molecular weight (~66 kDa), which is close to Lamin B1) and the unusual apparent molecular weight of SUN2 (~100 kDa versus the expected 80-85 kDa) would benefit from additional validation with a different antibody. The authors should also address the distorted shape of the Lamin B1 band in soft ECM lysates. This distortion is likely an electrophoresis artifact caused by the hydrogels absorbing BSA from the culture medium, which is then released during lysis near the molecular weight of Lamin B1.

Response: Thank you for pointing out these. Regarding the molecular weight of SUN2, first, as suggested, we validated our finding with a different antibody (Proteintech, 27556-1-AP), and it detected SUN2 with the same size of WB band (~100 kDa) as our original antibody (Abcam, ab124916) (**Fig. R1**). Second, we searched the literatures and found out that two types of molecular weight of SUN2 (either ~80 or ~100 kDa) were reported in WB results in the literatures (PMID37828059, PMID39317734, PMID38177122, PMID36322767). We figured out the different exhibition of SUN2 molecular weight were probably due to the different profile of protein ladders. To prove that, we performed new experiments and ran the same cell lysates on the same SDS-PAGE gel with two different protein ladders, one is our original Thermo Scientific ladder (Cat# 26616) and the other is Bio-Rad ladder (Cat# 1610374) (**Fig R1. A**). With side-by-side comparison carefully, we confirm that SUN2 band aligned with Thermo ladder exhibits molecular weight ~100 kDa, whereas ~80 kDa with Bio-Rad ladder, which are consistent with both of SUN2 antibodies (**Fig R1. B, C**). Additionally, two representative figures from literatures are included here for your information, of which SUN2 bands are detected at ~100kDa by using the same Abcam antibody (ab124916) as ours (**Fig R1. D, E**). This consistency further confirmed that the detected band of SUN2 (~100 kDa) aligned to Thermo ladder in our study are correct. We have included **Fig R1 B, C** into the revised manuscript (**Fig. EV2A, B**).

***Panels A, D and E redacted as they contain published data from Bio-Rad, #1610374; Thermo #266216; *Nat Commun.* 2023, 14(1): 6416. PMID: 37828059.; *EMBO Rep.* 2024, 25(11): 4728-4748. PMID: 39317734.

Figure R1. Validation of the molecular weight of SUN2 band. (A) Profiles of two protein ladders, Bio-Rad and Thermo. (B) SUN2 was detected by primary antibody from Abcam, which was used in this study. The band was aligned with two proteins ladder and displayed molecular weight differently. (C) Same as (B), but detected by primary antibody from Proteintech. (D) and (E) two representative figures from literatures to show SUN2 molecular weight at ~100 kDa.

Regarding the distorted shape of the Lamin B1 band specifically in soft ECM lysates, we thank the reviewer for proposing the highly plausible explanation that absorbed BSA from the culture medium could co-migrate and distort bands near 66 kDa. We performed two control experiments: (1) a nuclear fractionation assay to remove contaminating BSA from soft gel samples, and (2) direct spiking of BSA into stiff gel samples. The results confirm that the observed band distortion is attributable to BSA co-migration. Furthermore, the reduction in Lamin B1 on soft gels remains robust and reproducible in purified nuclear fractions, validating our central finding. We have incorporated the results into the revised manuscript (**Fig. EV2C-E**) (**Page 6, Lines 14-16**).

Figure EV2. Validation of the molecular weight of SUN2 band and the band shape of Lamin B1. (C) Lamin B1 protein levels and band shapes were examined with whole cell lysates (WCL) or nuclear fractions (Nucleus) from FN-coated 0.2 kPa or 10 kPa gels. (D) Statistical analysis of lamin B1 protein levels among the groups from (C). n= 3 biological replicates. Data are presented as the mean \pm SD. Tukey's multiple comparisons test. (E) Lamin B1 protein levels and band shapes were examined with or without BSA added. Protein samples were whole cell lysates from soft gels or stiff gels coated with FN.

Dear Prof. Peng

Thank you for the submission of your revised manuscript to EMBO reports. We have now received the full set of referee reports that is copied below.

As you can see, the referees find that the study is significantly improved during revision and recommend publication. Before I can accept the manuscript, I need you to address some editorial points below:

- As a standard procedure, we edit the title and abstract of manuscripts to make them more accessible to a general readership. Please find the suggested versions below my signature.

- The manuscript sections are in the wrong order. Please order them like this:

Title page - Abstract - Introduction - Results - Discussion - Methods - Acknowledgements - Disclosure and competing interests statement - References - Figure legends - Tables and their legends (not EV tables) - Expanded View Figure legends
All figures should be removed and provided separately and the legends of main and EV figures provided at the end of the manuscript.

- Please remove the Abbreviations section from the manuscript. Abbreviations should be defined in brackets after their first mention in the text.

- Please rename the "Materials and Methods" section to "Methods."

- Please move the description of power calculation and blinding to the Methods section.

- Please include a statement in the Methods section whether cell lines were authenticated.

- Please update the 'Conflict of interest' paragraph to our new 'Disclosure and competing interests statement'. For more information see

<https://www.embopress.org/page/journal/14693178/authorguide#conflictsofinterest>

In addition, employment in a biotech company should be stated in the Disclosure and competing interests statement

- Regarding the Author Contributions, we now use CRediT to specify the contributions of each author in the journal submission system. Therefore, please remove the Author Contributions from the manuscript file and make sure that the author contributions in our online manuscript tracking system are correct and up-to-date. The information you specified in the system will be automatically retrieved and typeset into the article. You can enter additional information in the free text box provided, if you wish.

- We request the ORCID ID number be linked to the account in our online system for Dr.s Quhuan Li and Juhui Qiu, as this is obligatory for all co-corresponding authors

- Please double-check to make sure to all relevant funding information in the manuscript is also entered into our submission system. (Missing in the system currently: Open Sharing Fund for the Large Instruments and Equipments of Shenzhen Bay Laboratory)

This needs to be added via the More Funders option (Comments Box should not be used).

- Please note that EMBO press papers are accompanied online by:

A) a short (2 sentences) summary of the findings and their significance,

B) 2-5 short bullet points highlighting the key results, and

C) a synopsis image in .jpg or .png format that is exactly 550 pixels wide and 300-600 pixels high (the height is variable). Please note that the text needs to be legible at the final size. Please upload this information along with your revised manuscript (the text for A and B should be provided in one separate Word file uploaded as Cover Art/Synopsis Image).

- Please remove the Reagent and Tools Table from the manuscript file and upload it as an individual file. For more information, please check <https://www.embopress.org/page/journal/14602075/authorguide#structuredmethods> and download the template for Reagent Table (attached for your convenience)

It should not be titles and called out as Table 1 but as Reagents and Tools Table.

- Please include the oligonucleotides used in this study in the Reagents and Tools Table. Alternatively, they may be placed in a separate table and that table referenced in the Reagents and Tools table.

- BioRender should be acknowledged at the end of the Methods section in the following way:

Graphics:

(some of the... OR Figure #... OR synopsis) Graphics were created with BioRender.com.

Additional information on source data and instruction on how to label the files are available
<https://www.embopress.org/page/journal/14693178/authorguide#sourcedata>

- Before submitting your revision, primary datasets (and computer code, where appropriate) produced in this study need to be deposited in an appropriate public database (see <
<https://www.embopress.org/page/journal/14693178/authorguide#dataavailability>>).
Specifically, we kindly request you provide public access to your RNA sequencing dataset.
The accession numbers and database should be listed in a formal "Data Availability " section (placed after Materials & Method) that follows the model below (see also <
<https://www.embopress.org/page/journal/14693178/authorguide#dataavailability>>).
Please note that the Data Availability Section is restricted to new primary data that are part of this study.

Data availability

- The resolution of the submitted figures is too low for blot and microscopy images. This reduction in resolution is commonly caused by converting original 16-bit TIFF files to RGB format for publication. While this is not inherently problematic, it can raise concerns about image integrity for critical readers.

To avoid any misunderstanding and to meet EMBO Press standards, we kindly ask that you:

* Resubmit the complete figure set at the captured original data resolution.

* Apply the same resolution standards to the source data files, which are also currently below the required quality threshold.

Please upload the blot source data files as .tiff or pdf.

-Additionally, please address these comments on the figure legends from our editorial team:

- Please note that the exact p values are not provided in the legends of figures 1E, G, H, K, O,P; 2E, 3I, Q, R; 4G, H, L; 5B, C, G, L, R, S, U; EV1 C, EV4 B, E, H, I

- Please indicate the statistical test used for data analysis in the legend of figure 1B

- Please note that information related to n is missing in the legend of figure 3C

- Although 'n' is provided, please describe the nature of entity for 'n' in the legends of figures 2C, G; 3E, G, I, L, N, O; 4C, E, J, P; 5O, R, S; EV1 C, EV3 B, EV4 B, D, M; EV5 D

- Please note that the arrow heads are not defined in the legend of figure 1M. This needs to be rectified.

With kind regards,

Kurt Weir
Editor
EMBO Reports

Cells decode mechanical cues to direct fate decisions through nuclear remodeling, yet nuclear adaptors to mechanical signals remain elusive. Here, we show that soft matrix suppresses myoblast differentiation and induces nuclear abnormality within 30 minutes, accompanied by a greater than 60% reduction in lamin B1 proteins levels. Mechanistically, midnolin interacts with lamin B1 and mediates ubiquitination-independent degradation of lamin B1 on soft matrix, through the Catch domain of midnolin engaging a β -strand within lamin B1's Ig-like domain. Functionally, moderate lamin B1 expression is essential for myoblast differentiation initiation, as its depletion either by siRNA or CRISPR knockout abolishes myogenic capacity. Our findings reveal that the midnolin-proteasome axis directly converts mechanical inputs into lineage commitment by triggering lamin B1 degradation, defining a novel nuclear mechano-adaptation pathway.

Referee #3:

The authors have adequately responded to the points raised in my last review.

The authors addressed the remaining editorial issues.

Prof. Qin Peng
Shenzhen Bay Laboratory
Institute of Systems and Physical Biology
Kelian Road
Shenzhen, Guangdong 518067
China

Dear Prof. Peng,

I am pleased to inform you that your manuscript has been accepted for publication in EMBO reports. Your manuscript will be processed for publication by EMBO Press. It will be copy edited and you will receive page proofs prior to publication. Please note that you will be contacted by Springer Nature Author Services to complete licensing and payment information.

You may qualify for financial assistance for your publication charges - either via a Springer Nature fully open access agreement or an EMBO initiative. Check your eligibility: <https://link.springer.com/journal/44319/how-to-publish-with-us>

Yours sincerely,

Kurt Weir
Editor
EMBO Reports

>>> Please note that it is EMBO Reports policy for the transcript of the editorial process (containing referee reports and your response letter) to be published as an online supplement to each paper. If you do NOT want this, you will need to inform the Editorial Office via email immediately. More information is available here: <https://link.springer.com/partners/embo-press/editorial-policies#Peer%20review>